# DIFFERENTIAL PRIVACY IN ADVERSARIAL LEARNING WITH PROVABLE ROBUSTNESS

## ABSTRACT

In this paper, we aim to develop a novel mechanism to preserve differential privacy (**DP**) in adversarial learning for deep neural networks, with *provable robustness to adversarial examples*. We leverage the sequential composition theory in DP, to establish a new connection between DP preservation and provable robustness. To address the trade-off among model utility, privacy loss, and robustness, we design an original, differentially private, adversarial objective function, based on the post-processing property in DP, to tighten the sensitivity of our model. An end-to-end theoretical analysis and thorough evaluations show that our mechanism notably improves the robustness of DP deep neural networks.

## 1 INTRODUCTION

The pervasiveness of machine learning exposes new vulnerabilities in software systems, in which deployed machine learning models can be used (a) to reveal sensitive information in private training data (Fredrikson et al., 2015), and/or (b) to make the models misclassify, such as *adversarial examples* (Carlini & Wagner, 2017). Efforts to prevent such attacks typically seek one of three solutions: **(1)** Models which preserve differential privacy (**DP**) (Dwork et al., 2006), a rigorous formulation of privacy in probabilistic terms; **(2)** Adversarial training algorithms, which augment training data to consist of benign examples and adversarial examples crafted during the training process, thereby empirically increasing the classification accuracy given adversarial examples (Kardan & Stanley, 2017; Matyasko & Chau, 2017); and **(3)** Provable robustness, in which the model classification given adversarial examples is theoretically guaranteed to be consistent, i.e., a small perturbation in the input does not change the predicted label (Cisse et al., 2017; Kolter & Wong, 2017).

On the one hand, *private models*, trained with existing privacy-preserving mechanisms (Abadi et al., 2016; Shokri & Shmatikov, 2015; Phan et al., 2016; 2017b;a; Yu et al., 2019; Lee & Kifer, 2018), are unshielded under adversarial examples. On the other hand, *robust models*, trained with adversarial learning algorithms (with or without provable robustness to adversarial examples), do not offer privacy protections to the training data. That one-sided approach poses serious risks to machine learning-based systems; since adversaries can attack a deployed model by using both privacy inference attacks and adversarial examples. To be safe, a model must be *i) private to protect the training data*, **and** *ii) robust to adversarial examples*. Unfortunately, there has not yet been research on how to develop such a model, which thus remains a largely open challenge.

Simply combining existing DP-preserving mechanisms and provable robustness conditions (Cisse et al., 2017; Kolter & Wong, 2017; Raghunathan et al., 2018) cannot solve the problem, for many reasons. **(a)** Existing sensitivity bounds (Phan et al., 2016; 2017b;a) and designs (Yu et al., 2019; Lee & Kifer, 2018) have not been developed to protect the training data in adversarial training. It is obvious that using adversarial examples crafted from the private training data to train our models introduces a previously unknown privacy risk, disclosing the participation of the benign examples (Song et al., 2019). **(b)** There is an unrevealed interplay among DP preservation, adversarial learning, and robustness bounds. **(c)** Existing algorithms cannot be readily applied to address the trade-off among model utility, privacy loss, and robustness. Therefore, theoretically bounding the robustness of a model (which both protects the privacy and is robust against adversarial examples) is nontrivial.

**Our Contributions.** Motivated by this open problem, we propose to develop a novel *differentially private adversarial learning* (**DPAL**) mechanism to: **1)** preserve DP of the training data, **2)** be provably and practically robust to adversarial examples, and **3)** retain high model utility. In our mech-

anism, privacy-preserving noise is injected into inputs and hidden layers to achieve DP in learning private model parameters (**Theorem 1**). Then, we incorporate ensemble adversarial learning into our mechanism to improve the decision boundary under DP protections. To do this, we introduce a concept of *DP adversarial examples* crafted using benign examples in the private training data under DP guarantees (**Eq. 9**). To address the trade-off between model utility and privacy loss, we propose a new DP adversarial objective function to tighten the model's global sensitivity (**Theorem 3**); thus, we significantly reduce the amount of noise injected into our function, compared with existing works (Phan et al., 2016; 2017b;a). In addition, *ensemble* DP adversarial examples with a dynamic perturbation size $\mu_a$ are introduced into the training process to further improve the robustness of our mechanism under different attack algorithms. An end-to-end privacy analysis shows that, by slitting the private training data into *disjoint* and *fixed* batches across epochs, the privacy budget in our DPAL is not accumulated across training steps (**Theorem 4**).

After preserving DP in learning model parameters, we establish a solid connection among privacy preservation, adversarial learning, and provable robustness. Noise injected into different layers is considered as a sequence of randomizing mechanisms, providing different levels of robustness. By leveraging the *sequential composition theory* in DP (Dwork & Roth, 2014), we derive a novel generalized robustness bound, which essentially is a composition of these levels of robustness (**Theorem 5** and **Proposition 1**). To our knowledge, our mechanism establishes the first connection between *DP preservation* and *provable robustness against adversarial examples* in *adversarial learning*. Rigorous experiments conducted on MNIST and CIFAR-10 datasets (Lecun et al., 1998; Krizhevsky & Hinton, 2009) show that our mechanism notably enhances the robustness of DP deep neural networks, compared with existing mechanisms.

## 2 BACKGROUND

In this section, we revisit adversarial learning, DP, and our problem definition. Let $D$ be a database that contains $N$ tuples, each of which contains data $x \in [-1, 1]^d$ and a *ground-truth label* $y \in \mathbb{Z}_K$, with $K$ possible categorical outcomes. Each $y$ is a one-hot vector of $K$ categories $y = \{y_1, \ldots, y_K\}$. A single *true class label* $y_x \in y$ given $x \in D$ is assigned to only one of the $K$ categories. On input $x$ and parameters $\theta$, a model outputs class scores $f : \mathbb{R}^d \to \mathbb{R}^K$ that maps $d$-dimensional inputs $x$ to a vector of scores $f(x) = \{f_1(x), \ldots, f_K(x)\}$ s.t. $\forall k \in [1, K] : f_k(x) \in [0, 1]$ and $\sum_{k=1}^{K} f_k(x) = 1$. The class with the highest score value is selected as the *predicted label* for the data tuple, denoted as $y(x) = \max_{k \in K} f_k(x)$. A loss function $L(f(x), y)$ presents the penalty for mismatching between the predicted values $f(x)$ and original values $y$. For the sake of clarity, the notations and terminologies frequently used in this paper are summarized in Table 1 (**Appendix A**). Let us briefly revisit DP-preserving techniques in deep learning, starting with the definition of DP.

**Definition 1** $(\epsilon, \delta)$-*DP (Dwork et al., 2006). A randomized algorithm A fulfills $(\epsilon, \delta)$-DP, if for any two databases $D$ and $D'$ differing at most one tuple, and for all $O \subseteq Range(A)$, we have:*

$$Pr[A(D) = O] \leq e^\epsilon Pr[A(D') = O] + \delta \tag{1}$$

*A smaller $\epsilon$ enforces a stronger privacy guarantee.*

Here, $\epsilon$ controls the amount by which the distributions induced by $D$ and $D'$ may differ, $\delta$ is a broken probability. DP also applies to general metrics $\rho(D, D') \leq 1$, where $\rho$ can be $l_p$-norms (Chatzikokolakis et al., 2013). DP-preserving algorithms in deep learning can be categorized into two lines: 1) introducing noise into *gradients* of parameters (Abadi et al., 2016; Shokri & Shmatikov, 2015; Abadi et al., 2017; Yu et al., 2019; Lee & Kifer, 2018; Phan et al., 2019), 2) injecting noise into objective functions (Phan et al., 2016; 2017b;a), and 3) injecting noise into labels (Papernot et al., 2018). In Lemmas 2 and 4, we will show that our mechanism achieves better sensitivity bounds compared with existing works (Phan et al., 2016; 2017b;a).

**Adversarial Learning.** For some target model $f$ and inputs $(x, y_x)$, the adversary's goal is to find an *adversarial example* $x^{\text{adv}} = x + \alpha$, where $\alpha$ is the perturbation introduced by the attacker, such that: **(1)** $x^{\text{adv}}$ and $x$ are close, and **(2)** the model misclassifies $x^{\text{adv}}$, i.e., $y(x^{\text{adv}}) \neq y(x)$. In this paper, we consider well-known $l_{p \in \{1, 2, \infty\}}$-norm bounded attacks (Goodfellow et al., 2014b). Let $l_p(\mu) = \{\alpha \in \mathbb{R}^d : \|\alpha\|_p \leq \mu\}$ be the $l_p$-norm ball of radius $\mu$. One of the goals in adversarial learning is to minimize the risk over adversarial examples: $\theta^* = \arg\min_\theta \mathbb{E}_{(x, y_{\text{true}}) \sim \mathcal{D}} \big[ \max_{\|\alpha\|_p \leq \mu} L\big(f(x + \alpha, \theta), y_x\big)\big]$, where an attack is used to approximate solutions to the inner maximization problem,

and the outer minimization problem corresponds to training the model $f$ with parameters $\theta$ over these adversarial examples $x^{\text{adv}} = x + \alpha$. There are two basic adversarial example attacks. The first one is a *single-step* algorithm, in which only a single gradient computation is required. For instance, **FGSM** algorithm (Goodfellow et al., 2014b) finds adversarial examples by solving the inner maximization $\max_{\|\alpha\|_p \leq \mu} L\big(f(x + \alpha, \theta), y_x\big)$. The second one is an *iterative* algorithm, in which multiple gradients are computed and updated. For instance, in (Kurakin et al., 2016a), FGSM is applied multiple times with $T_\mu$ small steps, each of which has a size of $\mu/T_\mu$.

To improve the robustness of models, prior work focused on two directions: 1) Producing correct predictions on adversarial examples, while not compromising the accuracy on legitimate inputs (Kardan & Stanley, 2017; Matyasko & Chau, 2017; Wang et al., 2016; Papernot et al., 2016b;a; Gu & Rigazio, 2014; Papernot & McDaniel, 2017; Hosseini et al., 2017); and 2) Detecting adversarial examples (Metzen et al., 2017; Grosse et al., 2017; Xu et al., 2017; Abbasi & Gagné, 2017; Gao et al., 2017). Among existing solutions, adversarial training appears to hold the greatest promise for learning robust models (Tramèr et al., 2017). One of the well-known algorithms was proposed in (Kurakin et al., 2016b). At every training step, new adversarial examples are generated and injected into batches containing both benign and adversarial examples. The typical adversarial learning in (Kurakin et al., 2016b) is presented in Alg. 2 (**Appendix B**).

**DP and Provable Robustness.** Recently, some algorithms (Cisse et al., 2017; Kolter & Wong, 2017; Raghunathan et al., 2018; Cohen et al., 2019; Li et al., 2018) have been proposed to derive provable robustness, in which each prediction is guaranteed to be consistent under the perturbation $\alpha$, if a robustness condition is held. Given a benign example $x$, we focus on achieving a robustness condition to attacks of $l_p(\mu)$-norm, as follows:

$$\forall \alpha \in l_p(\mu) : f_k(x + \alpha) > \max_{i:i \neq k} f_i(x + \alpha) \tag{2}$$

where $k = y(x)$, indicating that a small perturbation $\alpha$ in the input does not change the predicted label $y(x)$. To achieve the robustness condition in Eq. 2, Lecuyer et al. (Lecuyer et al., 2018) introduce an algorithm, called **PixelDP**. By considering an input $x$ (e.g., images) as databases in DP parlance, and individual features (e.g., pixels) as tuples, PixelDP shows that randomizing the scoring function $f(x)$ to enforce DP on a small number of pixels in an image guarantees robustness of predictions against adversarial examples. To randomize $f(x)$, *random noise $\sigma_r$* is injected into either input $x$ or an arbitrary hidden layer, resulting in the following $(\epsilon_r, \delta_r)$-PixelDP condition:

**Lemma 1** $(\epsilon_r, \delta_r)$-*PixelDP (Lecuyer et al., 2018). Given a randomized scoring function $f(x)$ satisfying $(\epsilon_r, \delta_r)$-PixelDP w.r.t. a $l_p$-norm metric, we have:*

$$\forall k, \forall \alpha \in l_p(1) : \mathbb{E}f_k(x) \leq e^{\epsilon_r}\mathbb{E}f_k(x + \alpha) + \delta_r \tag{3}$$

*where $\mathbb{E}f_k(x)$ is the expected value of $f_k(x)$, $\epsilon_r$ is a predefined budget, $\delta_r$ is a broken probability.*

At the prediction time, a certified robustness check is implemented for each prediction. A generalized robustness condition is proposed as follows:

$$\hat{\mathbb{E}}_{lb}f_k(x) > e^{2\epsilon_r} \max_{i:i \neq k} \hat{\mathbb{E}}_{ub}f_i(x) + (1 + e^{\epsilon_r})\delta_r \tag{4}$$

where $\hat{\mathbb{E}}_{lb}$ and $\hat{\mathbb{E}}_{ub}$ are the lower and upper bounds of the expected value $\hat{\mathbb{E}}f(x) = \frac{1}{n}\sum_n f(x)_n$, derived from the Monte Carlo estimation with an $\eta$-confidence, given $n$ is the number of invocations of $f(x)$ with independent draws in the noise $\sigma_r$. Passing the check for a given input guarantees that no perturbation up to $l_p(1)$-norm can change the model's prediction. PixelDP does not preserve DP in learning private parameters $\theta$ to protect the training data. That is different from our goal.

## 3 DPAL WITH PROVABLE ROBUSTNESS

Our new DPAL mechanism is presented in Alg. 1. Our network (Figure 1) can be represented as: $f(x) = g(a(x, \theta_1), \theta_2)$, where $a(x, \theta_1)$ is a feature representation learning model with $x$ as an input, and $g$ will take the output of $a(x, \theta_1)$ and return the class scores $f(x)$. At a high level, DPAL has three key components: **(1)** DP $a(x, \theta_1)$, which is to preserve DP in learning the feature representation model $a(x, \theta_1)$; **(2)** DP Adversarial Learning, which focuses on preserving DP in adversarial learning, given DP $a(x, \theta_1)$; and **(3)** Provable Robustness and Verified Inferring, which are to compute robustness bounds given an input at the inference time. In particular, given a deep neural network $f$ with model parameters $\theta$ (Lines 2-3), the network is trained over $T$ training steps. In each step, a batch of $m$ *perturbed* training examples and a batch of $m$ DP adversarial examples derived from $D$ are used to train our network (Lines 4-12).

---

**Algorithm 1 DPAL Mechanism**

---

**Input:** Database $D$, loss function $L$, parameters $\theta$, batch size $m$, learning rate $\varrho_t$, privacy budgets: $\epsilon_1$ and $\epsilon_2$, robustness parameters: $\epsilon_r$, $\Delta_r^x$, and $\Delta_r^h$, adversarial attack size $\mu_a$, the number of invocations $n$, ensemble attacks $A$, parameters $\psi$ and $\xi$, and the size $|\mathbf{h}_\pi|$ of $\mathbf{h}_\pi$

1: **Draw Noise** $\chi_1 \leftarrow [Lap(\frac{\Delta_\mathcal{R}}{\epsilon_1})]^d$, $\chi_2 \leftarrow [Lap(\frac{\Delta_\mathcal{R}}{\epsilon_1})]^\beta$, $\chi_3 \leftarrow [Lap(\frac{\Delta_{\mathcal{L}2}}{\epsilon_2})]^{|\mathbf{h}_\pi|}$

2: **Randomly Initialize** $\theta = \{\theta_1, \theta_2\}$, $\mathbf{B} = \{B_1, \ldots, B_{N/m}\}$ s.t. $\forall B \in \mathbf{B} : B$ is a batch with the size $m$, $B_1 \cap \ldots \cap B_{N/m} = \emptyset$, and $B_1 \cup \ldots \cup B_{N/m} = D$, $\overline{\mathbf{B}} = \{\overline{B}_1, \ldots, \overline{B}_{N/m}\}$ where $\forall i \in [1, N/m] : \overline{B}_i = \{\overline{x} \leftarrow x + \frac{\chi_1}{m}\}_{x \in B_i}$

3: **Construct** a deep network $f$ with **hidden layers** $\{\mathbf{h}_1 + \frac{2\chi_2}{m}, \ldots, \mathbf{h}_\pi\}$, where $\mathbf{h}_\pi$ is the last hidden layer

4: **for** $t \in [T]$ **do**

5:     **Take** a batch $\overline{B}_i \in \overline{\mathbf{B}}$ where $i = t\%(N/m)$, **Assign** $\overline{B}_t \leftarrow \overline{B}_i$

6:     **Ensemble DP Adversarial Examples:**

7:     **Draw Random Perturbation Value** $\mu_t \in (0, 1]$

8:     **Take** a batch $\overline{B}_{i+1} \in \overline{\mathbf{B}}$, **Assign** $\overline{B}_t^{\text{adv}} \leftarrow \emptyset$

9:     **for** $l \in A$ **do**

10:         **Take** the next batch $\overline{B}_a \subset \overline{B}_{i+1}$ with the size $m/|A|$

11:         $\forall \overline{x}_j \in \overline{B}_a$: **Craft** $\overline{x}_j^{\text{adv}}$ by using attack algorithm $A[l]$ with $l_\infty(\mu_t)$, $\overline{B}_t^{\text{adv}} \leftarrow \overline{B}_t^{\text{adv}} \cup \overline{x}_j^{\text{adv}}$

12:     **Descent:** $\theta_1 \leftarrow \theta_1 - \varrho_t \nabla_{\theta_1} \overline{\mathcal{R}}_{\overline{B}_t \cup \overline{B}_t^{\text{adv}}}(\theta_1)$; $\theta_2 \leftarrow \theta_2 - \varrho_t \nabla_{\theta_2} \overline{L}_{\overline{B}_t \cup \overline{B}_t^{\text{adv}}}(\theta_2)$ with the noise $\frac{\chi_3}{m}$

    **Output:** $(\epsilon_1 + \epsilon_1/\gamma_\mathbf{x} + \epsilon_1/\gamma + \epsilon_2)$-DP parameters $\theta = \{\theta_1, \theta_2\}$, robust model with an $\epsilon_r$ budget

13: **Verified Inferring:** (an input $x$, attack size $\mu_a$)

14: **Compute** robustness size $(\kappa + \varphi)_{max}$ in Eq. 15 of $x$

15: **if** $(\kappa + \varphi)_{max} \geq \mu_a$ **then**

16:     **Return** $isRobust(x) = True$, label $k$, $(\kappa + \varphi)_{max}$

17: **else**

18:     **Return** $isRobust(x) = False$, label $k$, $(\kappa + \varphi)_{max}$

---

## 3.1 DP Feature Representation Learning

Our idea is to use auto-encoder to simultaneously learn DP parameters $\theta_1$ and ensure that the output of $a(x, \theta_1)$ is DP. The reasons we choose an auto-encoder are: (1) It is easier to train, given its small size; and (2) It can be reused for different predictive models. A typical data reconstruction function (cross-entropy), given a batch $B_t$ at the training step $t$ of the input $x_i$, is as follows:

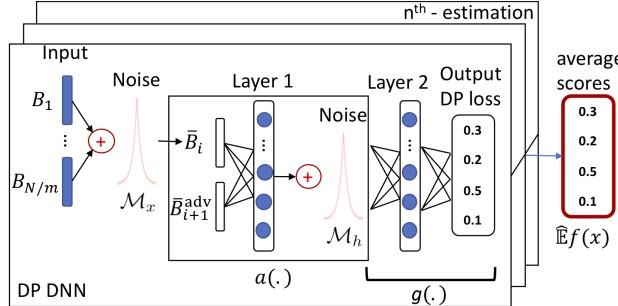

Figure 1: An instance of DPAL.

$$\mathcal{R}_{B_t}(\theta_1) = \sum_{x_i \in B_t} \sum_{j=1}^{d} \left[ x_{ij} \log(1 + e^{-\theta_{1j}h_i}) + (1 - x_{ij}) \log(1 + e^{\theta_{1j}h_i}) \right] \tag{5}$$

where the transformation of $x_i$ is $h_i = \theta_1^T x_i$, the hidden layer $\mathbf{h}_1$ of $a(x, \theta_1)$ given the batch $B_t$ is denoted as $\mathbf{h}_{1B_t} = \{\theta_1^T x_i\}_{x_i \in B_t}$, and $\widetilde{x}_i = \theta_1 h_i$ is the reconstruction of $x_i$.

To preserve $\epsilon_1$-DP in learning $\theta_1$ where $\epsilon_1$ is a privacy budget, we first derive the 1st-order polynomial approximation of $\mathcal{R}_{B_t}(\theta_1)$ by applying Taylor Expansion (Arfken, 1985), denoted as $\widetilde{\mathcal{R}}_{B_t}(\theta_1)$. Then, *Functional Mechanism* (Zhang et al., 2012) is employed to inject noise into coefficients of the approximated function $\widetilde{\mathcal{R}}_{B_t}(\theta_1) = \sum_{x_i \in B_t} \sum_{j=1}^{d} \sum_{l=1}^{2} \sum_{r=0}^{1} \frac{\mathbf{F}_{lj}^{(r)}(0)}{r!} (\theta_{1j}h_i)^r$, where $\mathbf{F}_{1j}(z) = x_{ij} \log(1 + e^{-z})$, $\mathbf{F}_{2j}(z) = (1 - x_{ij}) \log(1 + e^z)$, we have that: $\widetilde{\mathcal{R}}_{B_t}(\theta_1) = \sum_{x_i \in B_t} \sum_{j=1}^{d} \left[ \log 2 + \theta_{1j}(\frac{1}{2} - x_{ij})h_i \right]$. In $\widetilde{\mathcal{R}}_{B_t}(\theta_1)$, parameters $\theta_{1j}$ derived from the function optimization need to be $\epsilon_1$-DP. To achieve that, Laplace noise $\frac{1}{m}Lap(\frac{\Delta_\mathcal{R}}{\epsilon_1})$ is injected into coefficients $(\frac{1}{2} - x_{ij})h_i$, where $\Delta_\mathcal{R}$ is the sensitivity of $\widetilde{\mathcal{R}}_{B_t}(\theta_1)$, as follows:

$$\widetilde{\mathcal{R}}_{B_t}(\theta_1) = \sum_{x_i \in B_t} \sum_{j=1}^{d} \left[ \theta_{1j}\left( (\frac{1}{2} - x_{ij})h_i + \frac{1}{m}Lap(\frac{\Delta_\mathcal{R}}{\epsilon_1}) \right) \right] = \sum_{x_i \in B_t} \left[ \sum_{j=1}^{d} (\frac{1}{2}\theta_{1j}\overline{h}_i) - x_i\widetilde{x}_i \right] \tag{6}$$

To ensure that the computation of $\widetilde{x}_i$ does not access the original data, we further inject Laplace noise $\frac{1}{m} Lap(\frac{\Delta_\mathcal{R}}{\epsilon_1})$ into $x_i$. This can be done as a preprocessing step for all the benign examples in $D$ to construct a set of *disjoint* batches $\overline{\mathbf{B}}$ of perturbed benign examples (Lines 2 and 5). The perturbed function now becomes:

$$\overline{\mathcal{R}}_{\overline{B}_t}(\theta_1) = \sum_{\overline{x}_i \in \overline{B}_t} \Big[ \sum_{j=1}^{d} (\frac{1}{2}\theta_{1j}\overline{h}_i) - \overline{x}_i \widetilde{x}_i \Big] \tag{7}$$

where $\overline{x}_i = x_i + \frac{1}{m} Lap(\frac{\Delta_\mathcal{R}}{\epsilon_1})$, $h_i = \theta_1^T \overline{x}_i$, $\overline{h}_i = h_i + \frac{2}{m} Lap(\frac{\Delta_\mathcal{R}}{\epsilon_1})$, and $\widetilde{x}_i = \theta_1 \overline{h}_i$.

Let us denote $\beta$ as the number of neurons in $\mathbf{h}_1$, and $h_i$ is bounded in $[-1, 1]$, the global sensitivity $\Delta_\mathcal{R}$ is as follows:

**Lemma 2** *The global sensitivity of $\widetilde{\mathcal{R}}$ over any two neighboring batches, $B_t$ and $B_t'$, is as follows:* $\Delta_\mathcal{R} \leq d(\beta + 2)$.

All the proofs are in our **Appendix**. By setting $\Delta_\mathcal{R} = d(\beta + 2)$, we show that the output of $a(\cdot)$, which is the perturbed affine transformation $\overline{\mathbf{h}}_{1\overline{B}_t} = \{\overline{\theta}_1^T \overline{x}_i + \frac{2}{m} Lap(\frac{\Delta_\mathcal{R}}{\epsilon_1})\}_{\overline{x}_i \in \overline{B}_t}$, is $(\epsilon_1/\gamma)$-DP, given $\gamma = \frac{2\Delta_\mathcal{R}}{m\|\overline{\theta}_1\|_{1,1}}$ and $\|\overline{\theta}_1\|_{1,1}$ is the maximum 1-norm of $\theta_1$'s columns (Operator norm, 2018). This is important to tighten the privacy budget consumption in computing the remaining hidden layers $g(a(x, \theta_1), \theta_2)$. In fact, without using additional information from the original data, the computation of $g(a(x, \theta_1), \theta_2)$ is also $(\epsilon_1/\gamma)$-DP (the post-processing property of DP). Similarly, we observe that the perturbation of a batch $\overline{B}_t = \{\overline{x}_i \leftarrow x_i + \frac{\chi_1}{m}\}_{x_i \in B_t}$ achieves $(\epsilon_1/\gamma_\mathbf{x})$-DP, with $\gamma_\mathbf{x} = \frac{\Delta_\mathcal{R}}{m}$. Note that we do not use the post-processing property of DP to estimate the DP guarantee of $\overline{\mathbf{h}}_{1\overline{B}_t}$ based upon the DP guarantee of $\overline{B}_t$, since $\epsilon_1/\gamma < \epsilon_1/\gamma_\mathbf{x}$ in practice. As a result, the $(\epsilon_1/\gamma)$-DP $\overline{\mathbf{h}}_{1\overline{B}_t}$ provides a more rigorous DP protection to the computation of $g(\cdot)$ and to the output layer.

**Lemma 3** *The computation of the affine transformation $\overline{\mathbf{h}}_{1\overline{B}_t}$ is $(\epsilon_1/\gamma)$-DP and the computation of the batch $\overline{B}_t$ as the input layer is $(\epsilon_1/\gamma_\mathbf{x})$-DP.*

The following Theorem shows that optimizing $\overline{\mathcal{R}}_{\overline{B}_t}(\theta_1)$ is $(\epsilon_1/\gamma_\mathbf{x} + \epsilon_1)$-DP in learning $\theta_1$ given an $(\epsilon_1/\gamma_\mathbf{x})$-DP $\overline{B}_t$ batch.

**Theorem 1** *The optimization of $\overline{\mathcal{R}}_{\overline{B}_t}(\theta_1)$ preserves $(\epsilon_1/\gamma_\mathbf{x} + \epsilon_1)$-DP in learning $\theta_1$.*

### 3.2 DP ADVERSARIAL LEARNING

To integrate adversarial learning, we first draft DP adversarial examples $\overline{x}_j^{\text{adv}}$ using perturbed benign examples $\overline{x}_j$, with an ensemble of attack algorithms $A$ and a random perturbation budget $\mu_t \in (0, 1]$, at each step $t$ (Lines 6-11). This will significantly enhances the robustness of our models under different types of adversarial examples with an unknown adversarial attack size $\mu$.

$$\overline{x}_j^{\text{adv}} = \overline{x}_j + \mu \cdot \text{sign}\Big(\nabla_{\overline{x}_j}\mathcal{L}\big(f(\overline{x}_j, \theta), y(\overline{x}_j)\big)\Big) \tag{8}$$

with $y(\overline{x}_j)$ is the class prediction result of $f(\overline{x}_j)$ to avoid label leaking of the benign examples $x_j$ during the adversarial example crafting. Given a set of DP adversarial examples $\overline{B}_t^{\text{adv}}$, training the auto-encoder with $\overline{B}_t^{\text{adv}}$ preserves $(\epsilon_1/\gamma_\mathbf{x} + \epsilon_1)$-DP.

**Theorem 2** *The optimization of $\overline{\mathcal{R}}_{\overline{B}_t^{\text{adv}}}(\theta_1)$ preserves $(\epsilon_1/\gamma_\mathbf{x} + \epsilon_1)$-DP in learning $\theta_1$.*

The proof of Theorem 2 is in **Appendix H, Result 4**. It can be extended to iterative attacks as

$$\overline{x}_{j,0}^{\text{adv}} = \overline{x}_j, \overline{x}_{j,t+1}^{\text{adv}} = \overline{x}_{j,\text{t}}^{\text{adv}} + \frac{\mu}{T_\mu} \cdot \text{sign}\Big(\nabla_{\overline{x}_{j,\text{t}}^{\text{adv}}}\mathcal{L}\big(f(\overline{x}_{j,\text{t}}^{\text{adv}}, \theta), y(\overline{x}_{j,\text{t}}^{\text{adv}})\big)\Big) \tag{9}$$

where $y(\overline{x}_{j,\text{t}}^{\text{adv}})$ is the prediction result of $f(\overline{x}_{j,\text{t}}^{\text{adv}}, \theta)$, $\text{t} \in [0, T_\mu - 1]$.

Second, we propose a novel DP adversarial objective function $L_{B_t}(\theta_2)$, in which the loss function $\mathcal{L}$ for benign examples is combined with an additional loss function $\Upsilon$ for DP adversarial examples,

to optimize the parameters $\theta_2$. The objective function $L_{B_t}(\theta_2)$ is defined as follows:

$$L_{\overline{B}_t \cup \overline{B}_t^{\text{adv}}}(\theta_2) = \frac{1}{m(1+\xi)}\Big(\sum_{\overline{x}_i \in \overline{B}_t} \mathcal{L}\big(f(\overline{x}_i, \theta_2), y_i\big) + \xi \sum_{\overline{x}_j^{\text{adv}} \in \overline{B}_t^{\text{adv}}} \Upsilon\big(f(\overline{x}_j^{\text{adv}}, \theta_2), y_j\big)\Big) \quad (10)$$

where $\xi$ is a hyper-parameter. For the sake of clarity, in Eq. 10, we denote $y_i$ and $y_j$ as the true class labels $y_{x_i}$ and $y_{x_j}$ of examples $x_i$ and $x_j$. Note that $\overline{x}_j^{\text{adv}}$ and $x_j$ share the same label $y_{x_j}$.

Now we are ready to preserve DP in objective functions $\mathcal{L}\big(f(\overline{x}_i, \theta_2), y_i\big)$ and $\Upsilon\big(f(\overline{x}_j^{\text{adv}}, \theta_2), y_j\big)$ in order to achieve DP in learning $\theta_2$. Since the objective functions use the true class labels $y_i$ and $y_j$, we need to protect the labels at the output layer. Let us first present our approach to preserve DP in the objective function $\mathcal{L}$ for benign examples. Given $\mathbf{h}_{\pi i}$ computed from the $\overline{x}_i$ through the network with $W_\pi$ is the parameter at the last hidden layer $\mathbf{h}_\pi$. Cross-entropy function is approximated as

$$\mathcal{L}_{\overline{B}_t}(\theta_2) \cong \sum_{k=1}^{K} \sum_{\overline{x}_i} \Big[\mathbf{h}_{\pi i}W_{\pi k} - (\mathbf{h}_{\pi i}W_{\pi k})y_{ik} - \frac{1}{2}|\mathbf{h}_{\pi i}W_{\pi k}| + \frac{1}{8}(\mathbf{h}_{\pi i}W_{\pi k})^2\Big] \cong \mathcal{L}_{1\overline{B}_t}(\theta_2) - \mathcal{L}_{2\overline{B}_t}(\theta_2)$$

where $\mathcal{L}_{1\overline{B}_t}(\theta_2) = \sum_{k=1}^{K} \sum_{\overline{x}_i} \big[\mathbf{h}_{\pi i}W_{\pi k} - \frac{1}{2}|\mathbf{h}_{\pi i}W_{\pi k}| + \frac{1}{8}(\mathbf{h}_{\pi i}W_{\pi k})^2\big]$, and $\mathcal{L}_{2\overline{B}_t}(\theta_2) = \sum_{k=1}^{K} \sum_{\overline{x}_i}(\mathbf{h}_{\pi i}y_{ik})W_{\pi k}$. Based on the *post-processing property of DP* (Dwork & Roth, 2014), $\mathbf{h}_{\pi\overline{B}_t} = \{\mathbf{h}_{\pi i}\}_{\overline{x}_i \in \overline{B}_t}$ is $(\epsilon_1/\gamma)$-DP, since the computation of $\overline{\mathbf{h}}_{1\overline{B}_t}$ is $(\epsilon_1/\gamma)$-DP (Lemma 3). As a result, the optimization of the function $\mathcal{L}_{1\overline{B}_t}(\theta_2)$ does not disclose any information from the training data, and $\frac{Pr(\mathcal{L}_{1\overline{B}_t}(\theta_2))}{Pr(\mathcal{L}_{1\overline{B}_t'}(\theta_2))} = \frac{Pr(\mathbf{h}_{\pi\overline{B}_t})}{Pr(\mathbf{h}_{\pi\overline{B}_t'})} \leq e^{\epsilon_1/\gamma}$, given neighboring batches $\overline{B}_t$ and $\overline{B}_t'$. Thus, we only need to preserve $\epsilon_2$-DP in the function $\mathcal{L}_{2\overline{B}_t}(\theta_2)$, which accesses the ground-truth label $y_{ik}$. Given coefficients $\mathbf{h}_{\pi i}y_{ik}$, the sensitivity $\Delta_{\mathcal{L}2}$ of $\mathcal{L}_{2\overline{B}_t}(\theta_2)$ is computed as:

**Lemma 4** *Let $\overline{B}_t$ and $\overline{B}_t'$ be neighboring batches of benign examples, we have the following inequality: $\Delta_{\mathcal{L}2} \leq 2|\mathbf{h}_\pi|$, where $|\mathbf{h}_\pi|$ is the number of hidden neurons in $\mathbf{h}_\pi$.*

The sensitivity of our objective function is notably smaller than the state-of-the-art bound (Phan et al., 2017a), which is crucial to improve our model utility. The perturbed functions are as follows:

$$\overline{\mathcal{L}}_{\overline{B}_t}(\theta_2) = \mathcal{L}_{1\overline{B}_t}(\theta_2) - \overline{\mathcal{L}}_{2\overline{B}_t}(\theta_2), \text{ where } \overline{\mathcal{L}}_{2\overline{B}_t}(\theta_2) = \sum_{k=1}^{K} \sum_{\overline{x}_i} \big(\mathbf{h}_{\pi i}y_{ik} + \frac{1}{m}Lap(\frac{\Delta_{\mathcal{L}2}}{\epsilon_2})\big)W_{\pi k}$$

**Theorem 3** *Algorithm 1 preserves $(\epsilon_1/\gamma + \epsilon_2)$-DP in the optimization of $\overline{\mathcal{L}}_{\overline{B}}(\theta_2)$.*

We apply the same technique to preserve $(\epsilon_1/\gamma + \epsilon_2)$-DP in the optimization of the function $\Upsilon\big(f(\overline{x}_j^{\text{adv}}, \theta_2), y_j\big)$ over the DP adversarial examples $\overline{x}_j^{\text{adv}} \in \overline{B}_t^{\text{adv}}$. As the perturbed functions $\overline{\mathcal{L}}$ and $\overline{\Upsilon}$ are always optimized given two disjoint batches $\overline{B}_t$ and $\overline{B}_t^{\text{adv}}$, the privacy budget used to preserve DP in the adversarial objective function $L_{B_t}(\theta_2)$ is $(\epsilon_1/\gamma + \epsilon_2)$, following the *parallel composition* property of DP (Dwork & Roth, 2014). The total budget to learn private parameters $\overline{\theta} = \{\overline{\theta}_1, \overline{\theta}_2\} = \arg\min_{\{\theta_1, \theta_2\}}(\overline{\mathcal{R}}_{\overline{B}_t \cup \overline{B}_t^{\text{adv}}}(\theta_1) + \overline{L}_{\overline{B}_t \cup \overline{B}_t^{\text{adv}}}(\theta_2))$ is $(\epsilon_1 + \epsilon_1/\gamma + \epsilon_2)$ (Line 12).

We have shown that our mechanism achieves DP at the batch level $\overline{B}_t \cup \overline{B}_t^{\text{adv}}$ given a specific training step $t$. By constructing *disjoint* and *fixed* batches from the training data $D$, we leverage both parallel composition and post-processing properties of DP (Dwork & Roth, 2014) to extend the result to $(\epsilon_1 + \epsilon_1/\gamma_{\mathbf{x}} + \epsilon_1/\gamma + \epsilon_2)$-DP in learning $\overline{\theta} = \{\overline{\theta}_1, \overline{\theta}_2\}$ on $D$ across $T$ training steps. There are three key properties in our approach: **(1)** It only reads perturbed inputs $\overline{B}_t$ and perturbed coefficients $\overline{\mathbf{h}}_1$, which are DP across $T$ training steps; **(2)** Given $N/m$ disjoint batches in each epoch, for any example $\overline{x}$, $\overline{x}$ is included in *one and only one* batch, denoted $B_x \in \overline{\mathbf{B}}$. As a result, the DP guarantee to $\overline{x}$ in $D$ is equivalent to the DP guarantee to $\overline{x}$ in $B_x$; since the optimization using any other batches does not affect the DP guarantee of $\overline{x}$; and **(3)** All the batches are fixed across $T$ training steps to prevent additional privacy leakage, caused by generating new and overlapping batches (which are considered overlapping datasets in the parlance of DP) in the typical training approach.

**Theorem 4** *Algorithm 1 achieves $(\epsilon_1 + \epsilon_1/\gamma_{\mathbf{x}} + \epsilon_1/\gamma + \epsilon_2)$-DP parameters $\overline{\theta} = \{\overline{\theta}_1, \overline{\theta}_2\}$ on the private training data $D$ across $T$ training steps.*

### 3.3 PROVABLE ROBUSTNESS

Now, we establish the correlation between our mechanism and provable robustness. In the *inference time*, to derive the provable robustness condition against adversarial examples $x+\alpha$, i.e., $\forall \alpha \in l_p(1)$, PixelDP mechanism randomizes the scoring function $f(x)$ by injecting *robustness noise* $\sigma_r$ into either input $x$ or a hidden layer, i.e., $x' = x + Lap(\frac{\Delta_r^x}{\epsilon_r})$ or $h' = h + Lap(\frac{\Delta_r^h}{\epsilon_r})$, where $\Delta_r^x$ and $\Delta_r^h$ are the sensitivities of $x$ and $h$, measuring how much $x$ and $h$ can be changed given the perturbation $\alpha \in l_p(1)$ in the input $x$. Monte Carlo estimation of the expected values $\hat{\mathbb{E}}f(x)$, $\hat{\mathbb{E}}_{lb}f_k(x)$, and $\hat{\mathbb{E}}_{ub}f_k(x)$ are used to derive the robustness condition in Eq. 4.

On the other hand, in our mechanism, the privacy noise $\sigma_p$ includes Laplace noise injected into both input $x$, i.e., $\frac{1}{m}Lap(\frac{\Delta_{\mathcal{R}}}{\epsilon_1})$, and its affine transformation $h$, i.e., $\frac{2}{m}Lap(\frac{\Delta_{\mathcal{R}}}{\epsilon_1})$. Note that the perturbation of $\overline{\mathcal{L}}_{2\overline{B}_t}(\theta_2)$ is equivalent to $\overline{\mathcal{L}}_{2\overline{B}_t}(\theta_2) = \sum_{k=1}^{K}\sum_{\overline{x}_i}(\mathbf{h}_{\pi i}y_{ik}W_{\pi k} + \frac{1}{m}Lap(\frac{\Delta_{\mathcal{L}2}}{\epsilon_2})W_{\pi k})$. This helps us to avoid injecting the noise directly into the coefficients $\mathbf{h}_{\pi i}y_{ik}$. The correlation between our DP preservation and provable robustness lies in the correlation between the privacy noise $\sigma_p$ and the robustness noise $\sigma_r$.

*We can derive a robustness bound by projecting the privacy noise $\sigma_p$ on the scale of the robustness noise $\sigma_r$.* Given the input $x$, let $\kappa = \frac{\Delta_{\mathcal{R}}}{m\epsilon_1}/\frac{\Delta_r^x}{\epsilon_r}$, in our mechanism we have that: $\overline{x} = x + Lap(\kappa\Delta_r^x/\epsilon_r)$. By applying a group privacy size $\kappa$ (Dwork & Roth, 2014; Lecuyer et al., 2018), the scoring function $f(x)$ satisfies $\epsilon_r$-PixelDP given $\alpha \in l_p(\kappa)$, or equivalently is $\kappa\epsilon_r$-PixelDP given $\alpha \in l_p(1), \delta_r = 0$. By applying Lemma 1, we have

$$\forall k, \forall \alpha \in l_p(\kappa) : \mathbb{E}f_k(x) \leq e^{\epsilon_r}\mathbb{E}f_k(x+\alpha),$$
$$or \quad \forall k, \forall \alpha \in l_p(1) : \mathbb{E}f_k(x) \leq e^{(\kappa\epsilon_r)}\mathbb{E}f_k(x+\alpha) \tag{11}$$

With that, we can achieve a robustness condition against $l_p(\kappa)$-norm attacks, as follows:

$$\hat{\mathbb{E}}_{lb}f_k(x) > e^{2\epsilon_r}\max_{i:i\neq k}\hat{\mathbb{E}}_{ub}f_i(x) \tag{12}$$

with the probability $\geq \eta_x$-confidence, derived from the Monte Carlo estimation of $\hat{\mathbb{E}}f(x)$.

Our mechanism also perturbs $h$ (Eq. 7). Given $\varphi = \frac{2\Delta_{\mathcal{R}}}{m\epsilon_1}/\frac{\Delta_r^h}{\epsilon_r}$, we further have $\overline{h} = h + Lap(\frac{\varphi\Delta_r^h}{\epsilon_r})$. Therefore, the scoring function $f(x)$ also satisfies $\epsilon_r$-PixelDP given the perturbation $\alpha \in l_p(\varphi)$. In addition to the robustness to the $l_p(\kappa)$-norm attacks, we achieve an additional robustness bound in Eq. 12 against $l_p(\varphi)$-norm attacks. Similar to PixelDP, these robustness conditions can be achieved as randomization processes in the inference time. They can be considered as *two independent and provable defensive mechanisms* applied against two $l_p$-norm attacks, i.e., $l_p(\kappa)$ and $l_p(\varphi)$.

One challenging question here is: *"What is the general robustness bound, given $\kappa$ and $\varphi$?"* Intuitively, our model is robust to attacks with $\alpha \in l_p(\kappa + \varphi)$. We leverage the theory of *sequential composition* in DP (Dwork & Roth, 2014) to theoretically answer this question. Given $S$ independent mechanisms $\mathcal{M}_1, \ldots, \mathcal{M}_S$, whose privacy guarantees are $\epsilon_1, \ldots, \epsilon_S$-DP with $\alpha \in l_p(1)$. Each mechanism $\mathcal{M}_s$, which takes the input $x$ and outputs the value of $f(x)$ with the Laplace noise only injected to randomize the layer $s$ (i.e., no randomization at any other layers), denoted as $f^s(x)$, is defined as: $\forall s \in [1, S], \mathcal{M}_s f(x) : \mathbb{R}^d \rightarrow f^s(x) \in \mathbb{R}^K$. We aim to derive a generalized robustness of any composition scoring function $f(\mathcal{M}_1, \ldots, \mathcal{M}_s|x)$ bounded in $[0, 1]$, defined as follows:

$$f(\mathcal{M}_1, \ldots, \mathcal{M}_S|x) : \prod_{s=1}^{S}\mathcal{M}_s f(x) \Leftrightarrow f(\mathcal{M}_1, \ldots, \mathcal{M}_S|x) : \mathbb{R}^d \rightarrow \prod_{s=1}^{S}f^s(x) \in \mathbb{R}^K \tag{13}$$

Our setting follows the sequential composition in DP (Dwork & Roth, 2014). Thus, we can prove that the expected value $\mathbb{E}f(\mathcal{M}_1, \ldots, \mathcal{M}_S|x)$ is insensitive to small perturbations $\alpha \in l_p(1)$ in Lemma 5, and we derive our composition of robustness in Theorem 5, as follows:

**Lemma 5** *Given $S$ independent mechanisms $\mathcal{M}_1, \ldots, \mathcal{M}_S$, which are $\epsilon_1, \ldots, \epsilon_S$-DP w.r.t a $l_p$-norm metric, then the expected output value of any sequential function $f$ of them, i.e., $f(\mathcal{M}_1, \ldots, \mathcal{M}_S|x) \in [0, 1]$, meets the following property:*

$$\forall \alpha \in l_p(1) : \mathbb{E}f(\mathcal{M}_1, \ldots, \mathcal{M}_S|x) \leq e^{(\sum_{s=1}^{S}\epsilon_s)}\mathbb{E}f(\mathcal{M}_1, \ldots, \mathcal{M}_S|x+\alpha)$$

**Theorem 5** *(Composition of Robustness) Given $S$ independent mechanisms $\mathcal{M}_1, \ldots, \mathcal{M}_S$. Given any sequential function $f(\mathcal{M}_1, \ldots, \mathcal{M}_S | x)$, and let $\hat{\mathbb{E}}_{lb}$ and $\hat{\mathbb{E}}_{ub}$ are lower and upper bounds with an $\eta$-confidence, for the Monte Carlo estimation of $\hat{\mathbb{E}} f(\mathcal{M}_1, \ldots, \mathcal{M}_S | x) = \frac{1}{n} \sum_n f(\mathcal{M}_1, \ldots, \mathcal{M}_S | x)_n = \frac{1}{n} \sum_n (\prod_{s=1}^{S} f^s(x)_n)$. For any input $x$,*

$$\text{if } \exists k \in K : \hat{\mathbb{E}}_{lb} f_k(\mathcal{M}_1, \ldots, \mathcal{M}_S | x) > e^{2(\sum_{s=1}^{S} \epsilon_s)} \max_{i:i \neq k} \hat{\mathbb{E}}_{ub} f_i(\mathcal{M}_1, \ldots, \mathcal{M}_S | x), \quad (14)$$

*then the predicted label $k = \arg\max_k \hat{\mathbb{E}} f_k(\mathcal{M}_1, \ldots, \mathcal{M}_S | x)$, is robust to adversarial examples $x + \alpha$, $\forall \alpha \in l_p(1)$, with probability $\geq \eta$, by satisfying: $\hat{\mathbb{E}} f_k(\mathcal{M}_1, \ldots, \mathcal{M}_S | x + \alpha) > \max_{i:i \neq k} \hat{\mathbb{E}} f_i(\mathcal{M}_1, \ldots, \mathcal{M}_S | x + \alpha)$, which is the targeted robustness condition in Eq. 2.*

It is worth noting that there is no $\eta_s$-confidence for each mechanism $s$, since we do not estimate the expected value $\hat{\mathbb{E}} f^s(x)$ independently. To apply the composition of robustness in our mechanism, the noise injections into the input $x$ and its affine transformation $h$ can be considered as two mechanisms $\mathcal{M}_x$ and $\mathcal{M}_h$, sequentially applied as $(\mathcal{M}_h(x), \mathcal{M}_x(x))$. When $\mathcal{M}_h(x)$ is applied by invoking $f(x)$ with independent draws in the noise $\chi_2$, the noise $\chi_1$ injected into $x$ is fixed; and vice-versa. By applying group privacy (Dwork & Roth, 2014) with sizes $\kappa$ and $\varphi$, the scoring functions $f^x(x)$ and $f^h(x)$, given $\mathcal{M}_x$ and $\mathcal{M}_h$, are $\kappa \epsilon_r$-DP and $\varphi \epsilon_r$-DP given $\alpha \in l_p(1)$. With Theorem 5, we have a generalized bound as follows:

**Proposition 1** *(DPAL Robustness). For any input $x$, if $\exists k \in K : \hat{\mathbb{E}}_{lb} f_k(\mathcal{M}_h, \mathcal{M}_x | x) > e^{2\epsilon_r} \max_{i:i \neq k} \hat{\mathbb{E}}_{ub} f_i(\mathcal{M}_h, \mathcal{M}_x | x)$ (i.e., Eq. 14), then the predicted label $k$ of our function $f(\mathcal{M}_h, \mathcal{M}_x | x)$ is robust to perturbations $\alpha \in l_p(\kappa + \varphi)$ with the probability $\geq \eta$, by satisfying*

$$\forall \alpha \in l_p(\kappa + \varphi) : \hat{\mathbb{E}} f_k(\mathcal{M}_h, \mathcal{M}_x | x + \alpha) > \max_{i:i \neq k} \hat{\mathbb{E}} f_i(\mathcal{M}_h, \mathcal{M}_x | x + \alpha)$$

### 3.4 Training and Verified Inferring

Our model is trained similarly to training typical deep neural networks. Parameters $\theta_1$ and $\theta_2$ are independently updated by applying gradient descent (Line 12). Regarding the inference time, we implement a *verified inference procedure* as a post-processing step (Lines 13-18). Our verified inference returns a *robustness size guarantee* for each example $x$, which is the maximal value of $\kappa + \varphi$, for which the robustness condition in Proposition 1 holds. Maximizing $\kappa + \varphi$ is equivalent to maximizing the robustness epsilon $\epsilon_r$, which is the only parameter controlling the size of $\kappa + \varphi$; since, all the other hyper-parameters, i.e., $\Delta_{\mathcal{R}}$, $m$, $\epsilon_1$, $\epsilon_2$, $\theta_1$, $\theta_2$, $\Delta_r^x$, and $\Delta_r^h$ are fixed given a well-trained model $f(x)$:

$$(\kappa + \varphi)_{max} = \max_{\epsilon_r} \frac{\Delta_{\mathcal{R}} \epsilon_r}{m \epsilon_1} \left( \frac{1}{\Delta_r^x} + \frac{2}{\Delta_r^h} \right) \text{ s.t. } \hat{\mathbb{E}}_{lb} f_k(x) > e^{2\epsilon_r} \max_{i:i \neq k} \hat{\mathbb{E}}_{ub} f_i(x) \text{ (i.e., Eq. 14)} \quad (15)$$

The prediction on an example $x$ is robust to attacks up to $(\kappa + \varphi)_{max}$. The failure probability $1$-$\eta$ can be made arbitrarily small by increasing the number of invocations of $f(x)$, with independent draws in the noise. Similar to (Lecuyer et al., 2018), Hoeffding's inequality is applied to *bound* the approximation error in $\hat{\mathbb{E}} f_k(x)$ and to *search* for the robustness bound $(\kappa + \varphi)_{max}$. We use the following sensitivity bounds $\Delta_r^h = \beta \|\theta_1\|_\infty$ where $\|\theta_1\|_\infty$ is the maximum 1-norm of $\theta_1$'s rows, and $\Delta_r^x = \mu d$ for $l_\infty$ attacks. We also propose a new way to draw independent noise following the distribution of $\chi_1 + \frac{1}{m} Lap(0, \frac{\Delta_{\mathcal{R}}}{\epsilon_1} / \psi)$ for the input $x$ and $2\chi_2 + \frac{2}{m} Lap(0, \frac{\Delta_{\mathcal{R}}}{\epsilon_1} / \psi)$ for the transformation $h$, where $\chi_1$ and $\chi_2$ are the fixed noise used to train the network, and $\psi$ is a parameter to control the distribution shifts between training and inferring. This new Monte Carlo Estimation of $\hat{\mathbb{E}} f(x)$ works better without affecting the DP bounds and the robustness (**Appendix L**).

## 4 Experimental Results

We have conducted an extensive experiment on the MNIST and CIFAR-10 datasets. We consider the class of $l_\infty$-bounded adversaries to see whether our mechanism could retain high model utility, while providing strong DP guarantees and protections against adversarial examples.

**Baseline Approaches.** Our **DPAL** mechanism is evaluated in comparison with state-of-the-art mechanisms in: (1) DP-preserving algorithms in deep learning, i.e., **DP-SGD** (Abadi et al., 2016), **AdLM** (Phan et al., 2017a); in (2) Provable robustness, i.e., **PixelDP** (Lecuyer et al., 2018); and in

(3) DP-preserving algorithms with provable robustness, i.e., **SecureSGD** given heterogeneous noise (Phan et al., 2019), and **SecureSGD-AGM** (Phan et al., 2019) given the Analytic Gaussian Mechanism (AGM) (Balle & Wang, 2018). To preserve DP, DP-SGD injects random noise into gradients of parameters, while AdLM is a Functional Mechanism-based approach. PixelDP is one of the state-of-the-art mechanisms providing provable robustness using DP bounds. SecureSGD is a combination of PixelDP and DP-SGD with an advanced heterogeneous noise distribution; i.e., "more noise" is injected into "more vulnerable" latent features, to improve the robustness. The baseline models share the same design in our experiment. Four white-box attacks were used, including **FGSM**, **I-FGSM**, Momentum Iterative Method (**MIM**) (Dong et al., 2017), and **MadryEtAl** (Madry et al., 2018).

**Model Configuration.** It is important to note that $x \in [-1, 1]^d$ in our setting, which is different from a common setting, $x \in [0, 1]^d$. Thus, a given attack size $\mu_a = 0.3$ in the setting of $x \in [0, 1]^d$ is equivalent to an attack size $2\mu_a = 0.6$ in our setting. The reason for using $x \in [-1, 1]^d$ is to achieve better model utility, while retaining the same global sensitivities to preserve DP, compared with $x \in [0, 1]^d$. Our model configurations are in **Appendix M** and our approximation error bound analysis is presented in **Appendix N**. As in (Lecuyer et al., 2018), we apply two accuracy metrics:

$$conventional\ acc = \frac{\sum_{i=1}^{|test|} isCorrect(x_i)}{|test|}; \quad certified\ acc = \frac{\sum_{i=1}^{|test|} isCorrect(x_i)\ \&\ isRobust(x_i)}{|test|}$$

where $|test|$ is the number of test cases, $isCorrect(\cdot)$ returns 1 if the model makes a correct prediction (else, returns 0), and $isRobust(\cdot)$ returns 1 if the robustness size is larger than a given attack size $\mu_a$ (else, returns 0). Our task of validation focuses on shedding light into the interplay among model utility, privacy loss, and robustness bounds, by learning 1) the impact of the privacy budget $\epsilon_t = (\epsilon_1 + \epsilon_1/\gamma_{\mathbf{x}} + \epsilon_1/\gamma + \epsilon_2)$, and 2) the impact of attack sizes $\mu_a$. *All statistical tests are 2-tail t-tests.* All experimental Figures are in **Appendix O**.

**Results on the MNIST Dataset.** Figure 2 illustrates the conventional accuracy of each model as a function of the privacy budget $\epsilon_t$ on the MNIST dataset under $l_\infty(\mu_a)$-norm attacks, with $\mu_a = 0.2$ (a pretty strong attack). It is clear that our DPAL outperforms AdLM, DP-SGD, SecureSGD, and SecureSGD-AGM, in all cases, with $p < 1.32e-4$. On average, we register a 22.36% improvement over SecureSGD ($p < 1.32e-4$), a 46.84% improvement over SecureSGD-AGM ($p < 1.83e-6$), a 56.21% improvement over AdLM ($p < 2.05e-10$), and a 77.26% improvement over DP-SGD ($p < 5.20e-14$), given our DPAL mechanism. AdLM and DP-SGD achieve the worst conventional accuracies. There is no guarantee provided in AdLM and DP-SGD. Thus, the accuracy of the AdLM and DPSGD algorithms seem to show no effect against adversarial examples, when the privacy budget is varied. This is in contrast to our DPAL model, the SecureSGD model, and the SecureSGD-AGM model, whose accuracies are proportional to the privacy budget.

When the privacy budget $\epsilon_t = 0.2$ (a tight DP protection), there are significant drops, in terms of conventional accuracy, given the baseline approaches. By contrast, our DPAL mechanism only shows a small degradation in the conventional accuracy (6.89%, from 89.59% to 82.7%), compared with a 37% drop in SecureSGD (from 78.64% to 41.64%), and a 32.89% drop in SecureSGD-AGM (from 44.1% to 11.2%) on average, when the privacy budget $\epsilon_t$ goes from 2.0 to 0.2. At $\epsilon_t = 0.2$, our DPAL mechanism achieves 82.7%, compared with 11.2% and 41.64% correspondingly for SecureSGD-AGM and SecureSGD. This is an important result, showing the ability to offer tight DP protections under adversarial example attacks in our model, compared with existing algorithms.

• Figure 4 presents the conventional accuracy of each model as a function of the attack size $\mu_a$ on the MNIST dataset, under a strong DP guarantee, $\epsilon_t = 0.2$. It is clear that our DPAL mechanism outperforms the baseline approaches in all cases. On average, our DPAL model improves 44.91% over SecureSGD ($p < 7.43e-31$), a 61.13% over SecureSGD-AGM ($p < 2.56e-22$), a 52.21% over AdLM ($p < 2.81e-23$), and a 62.20% over DP-SGD ($p < 2.57e-22$). More importantly, our DPAL model is resistant to different adversarial example algorithms with different attack sizes. When $\mu_a \geq 0.2$, AdLM, DP-SGD, SecureSGD, and SecureSGD-AGM become defenseless. We further register significantly drops in terms of accuracy, when $\mu_a$ is increased from 0.05 (a weak attack) to 0.6 (a strong attack), i.e., $19.87\%$ on average given our DPAL, across all attacks, compared with 27.76% (AdLM), 29.79% (DP-SGD), 34.14% (SecureSGD-AGM), and 17.07% (SecureSGD).

• Figure 6 demonstrates the certified accuracy as a function of $\mu_a$. The privacy budget is set to 1.0, offering a reasonable privacy protection. In PixelDP, the construction attack bound $\epsilon_r$ is set to 0.1, which is a pretty reasonable defense. With (small perturbation) $\mu_a \leq 0.2$, PixelDP achieves better certified accuracies under all attacks; since PixelDP does not preserve DP to protect the training

data, compared with other models. Meanwhile, our DPAL model outperforms all the other models when $\mu_a \geq 0.3$, indicating a stronger defense to more aggressive attacks. More importantly, our DPAL has a consistent certified accuracy to different attacks given different attack sizes, compared with baseline approaches. In fact, when $\mu_a$ is increased from 0.05 to 0.6, our DPAL shows a small drop (11.88% on average, from $84.29\%(\mu_a = 0.05)$ to $72.41\%(\mu_a = 0.6)$), compared with a huge drop of the PixelDP, i.e., from $94.19\%(\mu_a = 0.05)$ to $9.08\%(\mu_a = 0.6)$ on average under I-FGSM, MIM, and MadryEtAl attacks, and to $77.47\%(\mu_a = 0.6)$ under FGSM attack. Similarly, we also register significant drops in terms of certified accuracy for SecureSGD (78.74%, from 86.74% to 7.99%) and SecureSGD-AGM (81.97%, from 87.23% to 5.26%) on average. This is promising.

Our key observations are as follows. **(1)** Incorporating ensemble adversarial learning into DP preservation, with tightened sensitivity bounds and a random perturbation size $\mu_t \in [0, 1]$ at each training step, does enhance the consistency, robustness, and accuracy of our model against different attack algorithms with different levels of perturbations. **(2)** Our DPAL model outperforms baseline algorithms, including both DP-preserving and non-private approaches, in terms of conventional accuracy and certified accuracy in most of the cases. It is clear that existing DP-preserving approaches have not been designed to withstand against adversarial examples.

**Results on the CIFAR-10 Dataset** further strengthen our observations. In Figure 3, our DPAL clearly outperforms baseline models in all cases ($p < 6.17e-9$), especially when the privacy budget is small ($\epsilon_t < 4$), yielding strong privacy protections. On average conventional accuracy, our DPAL mechanism has an improvement of 10.42% over SecureSGD ($p < 2.59e - 7$), an improvement of 14.08% over SecureSGD-AGM ($p < 5.03e - 9$), an improvement of 29.22% over AdLM ($p < 5.28e - 26$), and a 14.62% improvement over DP-SGD ($p < 4.31e - 9$). When the privacy budget is increased from 2 to 10, the conventional accuracy of our DPAL model increases from 42.02% to 46.76%, showing a 4.74% improvement on average. However, the conventional accuracy of our model under adversarial example attacks is still low, i.e., 44.22% on average given the privacy budget at 2.0. This opens a long-term research avenue to achieve better robustness under strong privacy guarantees in adversarial learning.

• The accuracy of our model is consistent given different attacks with different adversarial perturbations $\mu_a$ under a rigorous DP protection ($\epsilon_t = 2.0$), compared with baseline approaches (Figure 5). In fact, when the attack size $\mu_a$ increases from 0.05 to 0.5, the conventional accuracies of the baseline approaches are remarkably reduced, i.e., a drop of 25.26% on average given the most effective baseline approach, SecureSGD. Meanwhile, there is a much smaller degradation (4.79% on average) in terms of the conventional accuracy observed in our DPAL model. Our model also achieves better accuracies compared with baseline approaches in all cases ($p < 8.2e - 10$). Figure 7 further shows that our DPAL model is more accurate than baseline approaches (i.e., $\epsilon_r$ is set to 0.1 in PixelDP) in terms of certified accuracy in all cases, with a tight privacy budget set to 2.0 ($p < 2.04e - 18$). We register an improvement of 21.01% in our DPAL model given the certified accuracy over SecureSGD model, which is the most effective baseline approach ($p < 2.04e - 18$).

## 5 CONCLUSION

In this paper, we established a connection among DP preservation to protect the training data, adversarial learning, and provable robustness. A sequential composition robustness theory was introduced to generalize robustness given any sequential and bounded function of independent defensive mechanisms. An original DP-preserving mechanism was designed to address the trade-off among model utility, privacy loss, and robustness by tightening the global sensitivity bounds. A new Monte Carlo Estimation was proposed to improve and stabilize the estimation of the robustness bounds; thus improving the certified accuracy under adversarial example attacks.

However, there are several limitations. First, the accuracy of our model under adversarial example attacks is still very low. Second, the mechanism scalability is dependent on the model structures. Third, further study is needed to address the threats from adversarial examples crafted by unseen attack algorithms. Fourth, in this study, our goal is to illustrate the difficulties in providing DP protections to the training data in adversarial learning with robustness bounds. The problem is more challenging when working with complex and large networks, such as ResNet (He et al., 2015), VGG16 (Zhang et al., 2015), LSTM (Hochreiter & Schmidhuber, 1997), and GAN (Goodfellow et al., 2014a). Fifth, there can be alternative approaches to draft and to use DP adversarial examples. Addressing these limitations needs significant efforts from both research and practice communities.

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

## A    Notations and Terminologies

Table 1: Notations and Terminologies.

| | |
|---|---|
| $D$ and $x$ | Training data with benign examples $x \in [-1,1]^d$ |
| $y = \{y_1, \ldots, y_K\}$ | One-hot label vector of $K$ categories |
| $f : \mathbb{R}^d \to \mathbb{R}^K$ | Function/model $f$ that maps inputs $x$ to a vector of scores $f(x) = \{f_1(x), \ldots, f_K(x)\}$ |
| $y_x \in y$ | A single true class label of example $x$ |
| $y(x) = \max_{k \in K} f_k(x)$ | Predicted label for the example $x$ given the function $f$ |
| $x^{\mathrm{adv}} = x + \alpha$ | Adversarial example where $\alpha$ is the perturbation |
| $l_p(\mu) = \{\alpha \in \mathbb{R}^d : \|\alpha\|_p \le \mu\}$ | The $l_p$-norm ball of attack radius $\mu$ |
| $(\epsilon_r, \delta_r)$ | Robustness budget $\epsilon_r$ and broken probability $\delta_r$ |
| $\mathbb{E}f_k(x)$ | The expected value of $f_k(x)$ |
| $\hat{\mathbb{E}}_{lb}$ and $\hat{\mathbb{E}}_{ub}$ | Lower and upper bounds of the expected value $\hat{\mathbb{E}}f(x) = \frac{1}{n}\sum_n f(x)_n$ |
| $a(x, \theta_1)$ | Feature representation learning model with $x$ and parameters $\theta_1$ |
| $B_t$ | A batch of benign examples $x_i$ |
| $\mathcal{R}_{B_t}(\theta_1)$ | Data reconstruction function given $B_t$ in $a(x, \theta_1)$ |
| $\mathbf{h}_{1B_t} = \{\theta_1^T x_i\}_{x_i \in B_t}$ | The values of all hidden neurons in the hidden layer $\mathbf{h}_1$ of $a(x, \theta_1)$ given the batch $B_t$ |
| $\widetilde{\mathcal{R}}_{B_t}(\theta_1)$ and $\overline{\mathcal{R}}_{\overline{B}_t}(\theta_1)$ | Approximated and perturbed functions of $\mathcal{R}_{B_t}(\theta_1)$ |
| $\overline{x}_i$ and $\widetilde{x}_i$ | Perturbed and reconstructed inputs $x_i$ |
| $\Delta_{\mathcal{R}} = d(\beta + 2)$ | Sensitivity of the approximated function $\widetilde{\mathcal{R}}_{B_t}(\theta_1)$ |
| $\overline{\mathbf{h}}_{1B_t}$ | Perturbed affine transformation $\mathbf{h}_{1B_t}$ |
| $\overline{x}_j^{\mathrm{adv}} = x_j^{\mathrm{adv}} + \frac{1}{m} Lap(\frac{\Delta_{\mathcal{R}}}{\epsilon_1})$ | DP adversarial examples crafting from benign example $x_j$ |
| $\overline{B}_t$ and $\overline{B}_t^{\mathrm{adv}}$ | Sets of perturbed inputs $\overline{x}_i$ and DP adversarial examples $\overline{x}_j^{\mathrm{adv}}$ |
| $\mathcal{L}_{\overline{B}_t}(\theta_2)$ | Loss function of perturbed benign examples in $\overline{B}_t$, given $\theta_2$ |
| $\Upsilon\big(f(\overline{x}_j^{\mathrm{adv}}, \theta_2), y_j\big)$ | Loss function of DP adversarial examples $\overline{x}_j^{\mathrm{adv}}$, given $\theta_2$ |
| $\overline{\mathcal{L}}_{\overline{B}_t}(\theta_2)$ | DP loss function for perturbed benign examples $\overline{B}_t$ |
| $\mathcal{L}_{2\overline{B}_t}(\theta_2)$ | A part of the loss function $\mathcal{L}_{\overline{B}_t}(\theta_2)$ that needs to be DP |
| $f(\mathcal{M}_1, \ldots, \mathcal{M}_s | x)$ | Composition scoring function given independent randomizing mechanisms $\mathcal{M}_1, \ldots, \mathcal{M}_s$ |
| $\Delta_r^x$ and $\Delta_r^h$ | Sensitivities of $x$ and $h$, given the perturbation $\alpha \in l_p(1)$ |
| $(\epsilon_1 + \epsilon_1/\gamma_{\mathbf{x}} + \epsilon_1/\gamma + \epsilon_2)$ | Privacy budget to protect the training data $D$ |
| $(\kappa + \varphi)_{max}$ | Robustness size guarantee given an input $x$ at the inference time |

## B    Pseudo-code of Adversarial Training (Kurakin et al., 2016b)

Given a loss function:

$$L(\theta) = \frac{1}{m_1 + \xi m_2}\Big( \sum_{x_i \in B_t} \mathcal{L}\big(f(x_i, \theta), y_i\big) + \xi \sum_{x_j^{\mathrm{adv}} \in B_t^{\mathrm{adv}}} \Upsilon\big(f(x_j^{\mathrm{adv}}, \theta), y_j\big) \Big) \tag{16}$$

where $m_1$ and $m_2$ correspondingly are the numbers of examples in $B_t$ and $B_t^{\mathrm{adv}}$ at each training step.

---

**Algorithm 2 Adversarial Training (Kurakin et al., 2016b)**

---

**Input:** Database $D$, loss function $L$, parameters $\theta$, batch sizes $m_1$ and $m_2$, learning rate $\varrho_t$, parameter $\xi$

1: **Initialize** $\theta$ randomly
2: **for** $t \in [T]$ **do**
3:    **Take** a random batch $B_t$ with the size $m_1$, and a random batch $B_a$ with the size $m_2$
4:    **Craft** adversarial examples $B_t^{\mathrm{adv}} = \{x_j^{\mathrm{adv}}\}_{j \in [1, m_2]}$ from corresponding benign examples $x_j \in B_a$
5:    **Descent:** $\theta \leftarrow \theta - \varrho_t \nabla_\theta L(\theta)$

---

## C  PROOF OF LEMMA 2

**Proof 1** *Assume that $B_t$ and $B_t'$ differ in the last tuple, $x_m$ ($x_m'$). Then,*

$$\Delta_{\mathcal{R}} = \sum_{j=1}^{d}\Big[\|\sum_{x_i\in B_t}\frac{1}{2}h_i - \sum_{x_i'\in B_t'}\frac{1}{2}h_i'\|_1 + \|\sum_{x_i\in B_t}x_{ij} - \sum_{x_i'\in B_t'}x_{ij}'\|_1\Big]$$

$$\leq 2\max_{x_i}\sum_{j=1}^{d}(\|\frac{1}{2}h_i\|_1 + \|x_{ij}\|_1) \leq d(\beta + 2)$$

## D  PROOF OF LEMMA 3

**Proof 2** *Regarding the computation of $\mathbf{h}_{1\overline{B}_t} = \{\overline{\theta}_1^T\overline{x}_i\}_{\overline{x}_i\in\overline{B}_t}$, we can see that $h_i = \overline{\theta}_1^T\overline{x}_i$ is a linear function of $x$. The sensitivity of a function $h$ is defined as the maximum change in output, that can be generated by a change in the input (Lecuyer et al., 2018). Therefore, the global sensitivity of $\mathbf{h}_1$ can be computed as follows:*

$$\Delta_{\mathbf{h}_1} = \frac{\|\sum_{\overline{x}_i\in\overline{B}_t}\overline{\theta}_1^T\overline{x}_i - \sum_{\overline{x}_i'\in\overline{B}_t'}\overline{\theta}_1^T\overline{x}_i'\|_1}{\|\sum_{\overline{x}_i\in\overline{B}_t}\overline{x}_i - \sum_{\overline{x}_i'\in\overline{B}_t'}\overline{x}_i'\|_1} \leq \max_{x_i\in B_t}\frac{\|\overline{\theta}_1^T\overline{x}_i\|_1}{\|\overline{x}_i\|_1} \leq \|\overline{\theta}_1^T\|_{1,1}$$

*following matrix norms (Operator norm, 2018): $\|\overline{\theta}_1^T\|_{1,1}$ is the maximum 1-norm of $\theta_1$'s columns. By injecting Laplace noise $Lap(\frac{\Delta_{\mathbf{h}_1}}{\epsilon_1})$ into $\mathbf{h}_{1B_t}$, i.e., $\overline{\mathbf{h}}_{1\overline{B}_t} = \{\overline{\theta}_1^T\overline{x}_i + Lap(\frac{\Delta_{\mathbf{h}_1}}{\epsilon_1})\}_{\overline{x}_i\in\overline{B}_t}$, we can preserve $\epsilon_1$-DP in the computation of $\overline{\mathbf{h}}_{1\overline{B}_t}$. Let us set $\Delta_{\mathbf{h}_1} = \|\overline{\theta}_1^T\|_{1,1}$, $\gamma = \frac{2\Delta_{\mathcal{R}}}{m\Delta_{\mathbf{h}_1}}$, and $\chi_2$ drawn as a Laplace noise $[Lap(\frac{\Delta_{\mathcal{R}}}{\epsilon_1})]^{\beta}$, in our mechanism, the perturbed affine transformation $\overline{\mathbf{h}}_{1\overline{B}_t}$ is presented as:*

$$\overline{\mathbf{h}}_{1\overline{B}_t} = \{\overline{\theta}_1^T\overline{x}_i + \frac{2\chi_2}{m}\}_{\overline{x}_i\in\overline{B}_t} = \{\overline{\theta}_1^T\overline{x}_i + \frac{2}{m}[Lap(\frac{\Delta_{\mathcal{R}}}{\epsilon_1})]^{\beta}\}_{\overline{x}_i\in\overline{B}_t}$$

$$= \{\overline{\theta}_1^T\overline{x}_i + [Lap(\frac{\gamma\Delta_{\mathbf{h}_1}}{\epsilon_1})]^{\beta}\}_{\overline{x}_i\in\overline{B}_t} = \{\overline{\theta}_1^T\overline{x}_i + [Lap(\frac{\Delta_{\mathbf{h}_1}}{\epsilon_1/\gamma})]^{\beta}\}_{\overline{x}_i\in\overline{B}_t}$$

*This results in an $(\epsilon_1/\gamma)$-DP affine transformation $\overline{\mathbf{h}}_{1B_t} = \{\overline{\theta}_1^T\overline{x}_i + [Lap(\frac{\Delta_{\mathbf{h}_1}}{\epsilon_1/\gamma})]^{\beta}\}_{\overline{x}_i\in\overline{B}_t}$.*

*Similarly, the perturbed inputs $\overline{B}_t = \{\overline{x}_i\}_{\overline{x}_i\in\overline{B}_t} = \{x_i + \frac{\chi_1}{m}\}_{x_i\in B_t} = \{x_i + [Lap(\frac{\Delta_{\mathbf{x}}}{\epsilon_1/\gamma_{\mathbf{x}}})]^{d}\}_{x_i\in B_t}$, where $\Delta_{\mathbf{x}}$ is the sensitivity measuring the maximum change in the input layer that can be generated by a change in the batch $B_t$ and $\gamma_{\mathbf{x}} = \frac{\Delta_{\mathcal{R}}}{m\Delta_{\mathbf{x}}}$. Following (Lecuyer et al., 2018), $\Delta_{\mathbf{x}}$ can be computed as follows: $\Delta_{\mathbf{x}} = \frac{\|\sum_{x_i\in B_t}\overline{x}_i - \sum_{x_i'\in B_t'}\overline{x}_i'\|_1}{\|\sum_{x_i\in B_t}\overline{x}_i - \sum_{x_i'\in B_t'}\overline{x}_i'\|_1} = 1$. As a result, the computation of $\overline{B}_t$ is $(\epsilon_1/\gamma_{\mathbf{x}})$-DP.*

*Consequently, Lemma 3 does hold.*

## E  PROOF OF THEOREM 1

**Proof 3** *Given $\chi_1$ drawn as a Laplace noise $[Lap(\frac{\Delta_{\mathcal{R}}}{\epsilon_1})]^{d}$ and $\chi_2$ drawn as a Laplace noise $[Lap(\frac{\Delta_{\mathcal{R}}}{\epsilon_1})]^{\beta}$, the perturbation of the coefficient $\phi \in \Phi = \{\frac{1}{2}h_i, x_i\}$, denoted as $\overline{\phi}$, can be rewritten as follows:*

$$for\ \phi\in\{x_i\}: \overline{\phi} = \sum_{x_i\in B}(\phi_{x_i} + \frac{\chi_1}{m}) = \sum_{x_i\in B}\phi_{x_i} + \chi_1 = \sum_{x_i\in B}\phi_{x_i} + [Lap(\frac{\Delta_{\mathcal{R}}}{\epsilon_1})]^{d}$$

$$for\ \phi\in\{\frac{1}{2}h_i\}: \overline{\phi} = \sum_{x_i\in B}\frac{1}{2}(h_i + \frac{2\chi_2}{m}) = \sum_{x_i\in B}(\phi_{x_i} + \frac{\chi_2}{m})$$

$$= \sum_{x_i\in B}\phi_{x_i} + \chi_2 = \sum_{x_i\in B}\phi_{x_i} + [Lap(\frac{\Delta_{\mathcal{R}}}{\epsilon_1})]^{\beta}$$

*we have*

$$Pr\big(\overline{\mathcal{R}}_{\overline{B}_t}(\theta_1)\big) = \prod_{j=1}^{d} \prod_{\phi \in \Phi} \exp\big(-\frac{\epsilon_1 \|\sum_{x_i \in B_t} \phi_{x_i} - \overline{\phi}\|_1}{\Delta_{\mathcal{R}}}\big)$$

$\Delta_{\mathcal{R}}$ *is set to* $d(\beta + 2)$, *we have that:*

$$\frac{Pr\big(\overline{\mathcal{R}}_{\overline{B}_t}(\theta_1)\big)}{Pr\big(\overline{\mathcal{R}}_{\overline{B}'_t}(\theta_1)\big)} = \frac{\prod_{j=1}^{d} \prod_{\phi \in \Phi} \exp\big(-\frac{\epsilon_1 \|\sum_{x_i \in B_t} \phi_{x_i} - \overline{\phi}\|_1}{\Delta_{\mathcal{R}}}\big)}{\prod_{j=1}^{d} \prod_{\phi \in \Phi} \exp\big(-\frac{\epsilon_1 \|\sum_{x'_i \in B'_t} \phi_{x'_i} - \overline{\phi}\|_1}{\Delta_{\mathcal{R}}}\big)}$$

$$\leq \prod_{j=1}^{d} \prod_{\phi \in \Phi} \exp(\frac{\epsilon_1}{\Delta_{\mathcal{R}}} \big\| \sum_{x_i \in B_t} \phi_{x_i} - \sum_{x'_i \in B'_t} \phi_{x'_i} \big\|_1)$$

$$\leq \prod_{j=1}^{d} \prod_{\phi \in \Phi} \exp(\frac{\epsilon_1}{\Delta_{\mathcal{R}}} 2 \max_{x_i \in B_t} \big\|\phi_{x_i}\big\|_1) \leq \exp(\frac{\epsilon_1 d(\beta + 2)}{\Delta_{\mathcal{R}}}) = \exp(\epsilon_1) \qquad (17)$$

*Consequently, the computation of* $\overline{\mathcal{R}}_{\overline{B}_t}(\theta_1)$ *preserves* $\epsilon_1$-DP *in Alg. 1. In addition, the parameter optimization of* $\overline{\mathcal{R}}_{\overline{B}_t}(\theta_1)$ *only uses the perturbed data* $\overline{B}_t$, *which is* $(\epsilon_1/\gamma_{\mathbf{x}})$-DP *(Lemma 3), in the computations of* $h_i$, $\overline{h}_i$, $\widetilde{x}_i$, *parameter gradients, and gradient descents at each step. These operations do not access the original dataset* $B_t$; *therefore, they do not incur any additional information from the original data (the post-processing property in DP Dwork & Roth (2014)). As a result, the total privacy budget to learn the perturbed optimal parameters* $\overline{\theta}_1$ *in Alg. 1 is* $(\epsilon_1/\gamma_{\mathbf{x}} + \epsilon_1)$-DP.

## F    PROOF OF LEMMA 4

**Proof 4** *Assume that* $\overline{B}_t$ *and* $\overline{B}'_t$ *differ in the last tuple, and* $\overline{x}_m$ ($\overline{x}'_m$) *be the last tuple in* $\overline{B}_t$ ($\overline{B}'_t$), *we have that*

$$\Delta_{\mathcal{L}2} = \sum_{k=1}^{K} \big\| \sum_{\overline{x}_i \in \overline{B}_t} (\mathbf{h}_{\pi i} y_{ik}) - \sum_{\overline{x}'_i \in \overline{B}'_t} (\mathbf{h}'_{\pi i} y'_{ik}) \big\|_1 = \sum_{k=1}^{K} \big\| \mathbf{h}_{\pi m} y_{mk} - \mathbf{h}'_{\pi m} y'_{mk} \big\|_1$$

*Since* $y_{mk}$ *and* $y'_{mk}$ *are one-hot encoding, we have that* $\Delta_{\mathcal{L}2} \leq 2\max_{\overline{x}_i} \|\mathbf{h}_{\pi i}\|_1$. *Given* $\mathbf{h}_{\pi i} \in [-1, 1]$, *we have*

$$\Delta_{\mathcal{L}2} \leq 2|\mathbf{h}_{\pi}| \qquad (18)$$

*Lemma 4 does hold.*

## G    PROOF OF THEOREM 3

**Proof 5** *Let* $\overline{B}_t$ *and* $\overline{B}'_t$ *be neighboring batches of benign examples, and* $\chi_3$ *drawn as Laplace noise* $[Lap(\frac{\Delta_{\mathcal{L}2}}{\epsilon_2})]^{|\mathbf{h}_{\pi}|}$, *the perturbations of the coefficients* $\mathbf{h}_{\pi i} y_{ik}$ *can be rewritten as:*

$$\overline{\mathbf{h}}_{\pi i} \overline{y}_{ik} = \sum_{\overline{x}_i} (\mathbf{h}_{\pi i} y_{ik} + \frac{\chi_3}{m}) = \sum_{\overline{x}_i} (\mathbf{h}_{\pi i} y_{ik}) + [Lap(\frac{\Delta_{\mathcal{L}2}}{\epsilon_2})]^{|\mathbf{h}_{\pi}|}$$

*Since all the coefficients are perturbed, and given* $\Delta_{\mathcal{L}2} = 2|\mathbf{h}_{\pi}|$, *we have that*

$$\frac{Pr(\mathcal{L}_{\overline{B}_t}(\theta_2))}{Pr(\mathcal{L}_{\overline{B}'_t}(\theta_2))} = \frac{Pr(\mathcal{L}_{1\overline{B}_t}(\theta_2))}{Pr(\mathcal{L}_{1\overline{B}'_t}(\theta_2))} \times \frac{Pr(\overline{\mathcal{L}}_{2\overline{B}_t}(\theta_2))}{Pr(\overline{\mathcal{L}}_{2\overline{B}'_t}(\theta_2))}$$

$$\leq e^{\epsilon_1/\gamma} \sum_{k=1}^{K} \frac{\exp(-\frac{\epsilon_2 \|\sum_{\overline{x}_i} \mathbf{h}_{\pi i} y_{ik} - \overline{\mathbf{h}}_{\pi i} \overline{y}_{ik}\|_1}{\Delta_{\mathcal{L}2}})}{\exp(-\frac{\epsilon_2 \|\sum_{\overline{x}'_i} \mathbf{h}_{\pi i} y_{ik} - \overline{\mathbf{h}}_{\pi i} \overline{y}_{ik}\|_1}{\Delta_{\mathcal{L}2}})}$$

$$\leq e^{\epsilon_1/\gamma} \sum_{k=1}^{K} \exp(\frac{\epsilon_2}{\Delta_{\mathcal{L}2}} \big\| \sum_{\overline{x}_i} \mathbf{h}_{\pi i} y_{ik} - \sum_{\overline{x}'_i} \mathbf{h}_{\pi i} y_{ik} \big\|_1)$$

$$\leq e^{\epsilon_1/\gamma} \exp(\frac{\epsilon_2}{\Delta_{\mathcal{L}2}} 2 \max_{\overline{x}_i} \|\mathbf{h}_{\pi i}\|_1) = e^{\epsilon_1/\gamma + \epsilon_2}$$

The computation of $\overline{\mathcal{L}}_{2\overline{B}_t}(\theta_2)$ preserves $(\epsilon_1/\gamma + \epsilon_2)$-differential privacy. The optimization of $\overline{\mathcal{L}}_{2\overline{B}_t}(\theta_2)$ does not access additional information from the original input $x_i \in B_t$. Consequently, the optimal perturbed parameters $\overline{\theta}_2$ derived from $\overline{\mathcal{L}}_{2\overline{B}_t}(\theta_2)$ are $(\epsilon_1/\gamma + \epsilon_2)$-DP.

## H    PROOF OF THEOREM 4

**Proof 6** *First, we optimize for a single draw of noise during training (Line 3) and all the batches of perturbed benign examples are disjoint and fixed across epochs. As a result, the computation of $\overline{x}_i$ is equivalent to a data preprocessing step with DP, which does not incur any additional privacy budget consumption over $T$ training steps (the post-processing property of DP) (**Result 1**). That is different from repeatedly applying a DP mechanism on either the same or overlapping datasets causing the accumulation of the privacy budget.*

*Now, we show that our algorithm achieves DP at the dataset level $D$. Let us consider the computation of the first hidden layer, given any two neighboring datasets $D$ and $D'$ differing at most one tuple $\overline{x}_e \in D$ and $\overline{x}'_e \in D'$. For any $O = \prod_{i=1}^{N/m} o_i \in \prod_{i=1}^{N/m} \overline{\mathbf{h}}_{1\overline{B}_i}(\in \mathbb{R}^{\beta \times m})$, we have that*

$$\frac{P(\overline{\mathbf{h}}_{1D} = O)}{P(\overline{\mathbf{h}}_{1D'} = O)} = \frac{P(\overline{\mathbf{h}}_{1\overline{B}_1} = o_1)\ldots P(\overline{\mathbf{h}}_{1\overline{B}_{N/m}} = o_{N/m})}{P(\overline{\mathbf{h}}_{1\overline{B}'_1} = o_1)\ldots P(\overline{\mathbf{h}}_{1\overline{B}'_{N/m}} = o_{N/m})} \tag{19}$$

*By having disjoint and fixed batches, we have that:*

$$\exists! \tilde{B} \in \overline{\mathbf{B}} \text{ s.t. } x_e \in \tilde{B} \text{ and } \exists! \tilde{B}' \in \overline{\mathbf{B}}' \text{ s.t. } x'_e \in \tilde{B}' \tag{20}$$

*From Eqs. 19, 20, and Lemma 3, we have that*

$$\forall \overline{B} \in \mathbf{B}, \overline{B} \neq \tilde{B} : \overline{B} = \overline{B}' \Rightarrow \frac{P(\overline{\mathbf{h}}_{1\overline{B}} = o)}{P(\overline{\mathbf{h}}_{1\overline{B}'} = o)} = 1 \tag{21}$$

$$\text{Eqs. 20 and 21} \Rightarrow \frac{P(\overline{\mathbf{h}}_{1D} = O)}{P(\overline{\mathbf{h}}_{1D'} = O)} = \frac{P(\overline{\mathbf{h}}_{1\tilde{B}} = \tilde{o})}{P(\overline{\mathbf{h}}_{1\tilde{B}'} = \tilde{o})} \leq e^{\epsilon_1/\gamma} \tag{22}$$

*As a result, the computation of $\overline{\mathbf{h}}_{1D}$ is $(\epsilon_1/\gamma)$-DP given the data $D$, since the Eq. 22 does hold for any tuple $x_e \in D$. That is consistent with the parallel composition property of DP, in which batches can be considered disjoint datasets given $\overline{\mathbf{h}}_{1\overline{B}}$ as a DP mechanism (Dwork & Roth, 2014).*

*This does hold across epochs, since batches $\overline{\mathbf{B}}$ are disjoint and fixed among epochs. At each training step $t \in [1, T]$, the computation of $\overline{\mathbf{h}}_{1\overline{B}_t}$ does not access the original data. It only reads the perturbed batch of inputs $\overline{B}_t$, which is $(\epsilon_1/\gamma_{\mathbf{x}})$-DP (Lemma 3). Following the post-processing property in DP (Dwork & Roth, 2014), the computation of $\overline{\mathbf{h}}_{1\overline{B}_t}$ does not incur any additional information from the original data across $T$ training steps. (**Result 2**)*

*Similarly, we show that the optimization of the function $\overline{\mathcal{R}}_{\overline{B}_t}(\theta_1)$ is $(\epsilon_1/\gamma_{\mathbf{x}} + \epsilon_1)$-DP across $T$ training steps. As in Theorem 1 and Proof 3, we have that $Pr(\overline{\mathcal{R}}_{\overline{B}}(\theta_1)) = \prod_{j=1}^{d} \prod_{\phi \in \Phi} \exp\big(-\frac{\epsilon_1\|\sum_{x_i \in B} \phi_{x_i} - \overline{\phi}\|_1}{\Delta_{\mathcal{R}}}\big)$, where $\overline{B} \in \overline{\mathbf{B}}$. Given any two perturbed neighboring datasets $\overline{D}$ and $\overline{D}'$ differing at most one tuple $\overline{x}_e \in \overline{D}$ and $\overline{x}'_e \in \overline{D}'$:*

$$\frac{Pr(\overline{\mathcal{R}}_{\overline{D}}(\theta_1))}{Pr(\overline{\mathcal{R}}_{\overline{D}'}(\theta_1))} = \frac{Pr(\overline{\mathcal{R}}_{\overline{B}_1}(\theta_1))\ldots Pr(\overline{\mathcal{R}}_{\overline{B}_{N/m}}(\theta_1))}{Pr(\overline{\mathcal{R}}_{\overline{B}'_1}(\theta_1))\ldots Pr(\overline{\mathcal{R}}_{\overline{B}'_{N/m}}(\theta_1))} \tag{23}$$

*From Eqs. 20, 23, and Theorem 1, we have that*

$$\forall \overline{B} \in \mathbf{B}, \overline{B} \neq \tilde{B} : \overline{B} = \overline{B}' \Rightarrow \frac{P(\overline{\mathcal{R}}_{\overline{B}}(\theta_1))}{P(\overline{\mathcal{R}}_{\overline{B}'}(\theta_1))} = 1 \tag{24}$$

$$\text{Eqs. 23 and 24} \Rightarrow \frac{P(\overline{\mathcal{R}}_{\overline{D}}(\theta_1))}{P(\overline{\mathcal{R}}_{\overline{D}'}(\theta_1))} = \frac{P(\overline{\mathcal{R}}_{\tilde{B}}(\theta_1))}{P(\overline{\mathcal{R}}_{\tilde{B}'}(\theta_1))} \leq e^{\epsilon_1} \tag{25}$$

As a result, the optimization of $\overline{\mathcal{R}}_{\overline{D}}(\theta_1)$ is $(\epsilon_1/\gamma_{\mathbf{x}} + \epsilon_1)$-DP given the data $\overline{D}$ (which is $\epsilon_1/\gamma_{\mathbf{x}}$-DP (Lemma 3)), since the Eq. 25 does hold for any tuple $\overline{x}_e \in \overline{D}$. This is consistent with the parallel composition property in DP (Dwork & Roth, 2014), in which batches can be considered disjoint datasets and the optimization of the function on one batch does not affect the privacy guarantee in any other batch. In addition, $\forall t \in [1, T]$, the optimization of $\overline{\mathcal{R}}_{\overline{B}_t}(\theta_1)$ does not use any additional information from the original data $D$. Consequently, the privacy budget is $(\epsilon_1/\gamma_{\mathbf{x}} + \epsilon_1)$ across $T$ training steps, following the post-processing property in DP (Dwork & Roth, 2014) (**Result 3**).

Similarly, we can also prove that optimizing the data reconstruction function $\overline{\mathcal{R}}_{\overline{B}_t^{adv}}(\theta_1)$ given the DP adversarial examples crafted in Eqs. 8 and 9, i.e., $\overline{x}_j^{adv}$, is also $(\epsilon_1/\gamma_{\mathbf{x}} + \epsilon_1)$-DP given $t \in [1, T]$ on the training data $D$. First, DP adversarial examples $\overline{x}_j^{adv}$ are crafted from perturbed benign examples $\overline{x}_j$. As a result, the computation of the batch $\overline{B}_t^{adv}$ of DP adversarial examples is 1) $(\epsilon_1/\gamma_{\mathbf{x}})$-DP (the post-processing property of DP (Dwork & Roth, 2014)), and 2) does not access the original data $\forall t \in [1, T]$. In addition, the computation of $\overline{\mathbf{h}}_{1\overline{B}_t^{adv}}$ and the optimization of $\overline{\mathcal{R}}_{\overline{B}_t^{adv}}(\theta_1)$ correspondingly are $\epsilon_1/\gamma$-DP and $\epsilon_1$-DP. In fact, the data reconstruction function $\overline{\mathcal{R}}_{\overline{B}_t^{adv}}$ is presented as follows:

$$\overline{\mathcal{R}}_{\overline{B}_t^{adv}}(\theta_1) = \sum_{\overline{x}_j^{adv} \in \overline{B}_t^{adv}} \Big[ \sum_{i=1}^{d} (\frac{1}{2}\theta_{1i}\overline{h}_j^{adv}) - \overline{x}_j^{adv}\widetilde{x}_j^{adv} \Big]$$

$$= \sum_{\overline{x}_j^{adv} \in \overline{B}_t^{adv}} \Big[ \sum_{i=1}^{d} (\frac{1}{2}\theta_{1i}\overline{h}_j^{adv}) - \overline{x}_j\widetilde{x}_j^{adv} - \mu \cdot sign\Big(\nabla_{\overline{x}_j}\mathcal{L}\big(f(\overline{x}_j, \theta), y(\overline{x}_j)\big)\Big)\widetilde{x}_j^{adv} \Big]$$

$$= \sum_{\overline{x}_j^{adv} \in \overline{B}_t^{adv}} \Big[ \sum_{i=1}^{d} (\frac{1}{2}\theta_{1i}\overline{h}_j^{adv}) - \overline{x}_j\widetilde{x}_j^{adv} \Big] - \sum_{\overline{x}_j^{adv} \in \overline{B}_t^{adv}} \mu \cdot sign\Big(\nabla_{\overline{x}_j}\mathcal{L}\big(f(\overline{x}_j, \theta), y(\overline{x}_j)\big)\Big)\widetilde{x}_j^{adv} \quad (26)$$

where $h_j^{adv} = \theta_1^T \overline{x}_j^{adv}$, $\overline{h}_j^{adv} = h_j^{adv} + \frac{2}{m}Lap(\frac{\Delta_{\mathcal{R}}}{\epsilon_1})$, and $\widetilde{x}_j^{adv} = \theta_1 \overline{h}_j^{adv}$. The right summation component in Eq. 26 does not disclose any additional information, since the $sign(\cdot)$ function is computed from perturbed benign examples (the post-processing property in DP (Dwork & Roth, 2014)). Meanwhile, the left summation component has the same form with $\overline{\mathcal{R}}_{\overline{B}_t}(\theta_1)$ in Eq. 7. Therefore, we can employ the Proof 3 in Theorem 1, by replacing the coefficients $\Phi = \{\frac{1}{2}h_i, x_i\}$ with $\Phi = \{\frac{1}{2}h_j^{adv}, x_j\}$ to prove that the optimization of $\overline{\mathcal{R}}_{\overline{B}_t^{adv}}(\theta_1)$ is $(\epsilon_1/\gamma_{\mathbf{x}} + \epsilon_1)$-DP. As a result, Theorem 2 does hold. (**Result 4**)

In addition to the Result 4, by applying the same analysis in Result 3, we can further show that the optimization of $\overline{\mathcal{R}}_{D^{adv}}(\theta_1)$ is $(\epsilon_1/\gamma_{\mathbf{x}} + \epsilon_1)$-DP given the DP adversarial examples $D^{adv}$ crafted using the data $\overline{D}$ across $T$ training steps, since batches used to created DP adversarial examples are disjoint and fixed across epochs. It is also straightforward to conduct the same analysis in Result 2, in order to prove that the computation of the first affine transformation $\overline{\mathbf{h}}_{1\overline{B}_t^{adv}} = \{\overline{\theta}_1^T \overline{x}_j^{adv} + \frac{2}{m}Lap(\frac{\Delta_{\mathcal{R}}}{\epsilon_1})\}_{\overline{x}_j^{adv} \in \overline{B}_t^{adv}}$ given the batch of DP adversarial examples $\overline{B}_t^{adv}$, is $(\epsilon_1/\gamma)$-DP with $t \in [1, T]$ training steps. This is also true given the data level $D^{adv}$. (**Result 5**)

Regarding the output layer, the Algorithm 1 preserves $(\epsilon_1/\gamma + \epsilon_2)$-DP in optimizing the adversarial objective function $\overline{L}_{\overline{B}_t \cup \overline{B}_t^{adv}}(\theta_2)$ (Theorem 3). We apply the same technique to preserve $(\epsilon_1/\gamma + \epsilon_2)$-DP across $T$ training steps given disjoint and fixed batches derived from the private training data $D$. In addition, as our objective functions $\overline{\mathcal{R}}$ and $\overline{L}$ are always optimized given two disjoint batches $\overline{B}_t$ and $\overline{B}_t^{adv}$, the privacy budget used to preserve DP in these functions is $(\epsilon_1 + \epsilon_1/\gamma + \epsilon_2)$, following the parallel composition property in DP (Dwork & Roth, 2014). (**Result 6**)

With the **Results 1-6**, all the computations and optimizations in the Algorithm 1 are DP following the post-processing property in DP (Dwork & Roth, 2014), by working on perturbed inputs and perturbed coefficients. The crafting and utilizing processes of DP adversarial examples based on the perturbed benign examples do not disclose any additional information. The optimization of

*our DP adversarial objective function at the output layer is DP to protect the ground-truth labels. More importantly, the DP guarantee in learning given the whole dataset level $\overline{D}$ is equivalent to the DP guarantee in learning on disjoint and fixed batches across epochs. Consequently, Algorithm 1 preserves $(\epsilon_1 + \epsilon_1/\gamma_{\mathbf{x}} + \epsilon_1/\gamma + \epsilon_2)$-DP in learning private parameters $\overline{\theta} = \{\overline{\theta}_1, \overline{\theta}_2\}$ given the training data $D$ across $T$ training steps. Note that the $\epsilon_1/\gamma_{\mathbf{x}}$ is counted for the perturbation on the benign examples. Theorem 4 does hold.*

## I    PROOF OF LEMMA 5

**Proof 7** *Thanks to the sequential composition theory in DP (Dwork & Roth, 2014), $f(\mathcal{M}_1, \ldots, \mathcal{M}_S | x)$ is $(\sum_s \epsilon_s)$-DP, since for any $O = \prod_{s=1}^{S} o_s \in \prod_{s=1}^{S} f^s(x) (\in \mathbb{R}^K)$, we have that*

$$\frac{P\big(f(\mathcal{M}_1, \ldots, \mathcal{M}_S | x) = O\big)}{P\big(f(\mathcal{M}_1, \ldots, \mathcal{M}_S | x + \alpha) = O\big)} = \frac{P(\mathcal{M}_1 f(x) = o_1) \ldots P(\mathcal{M}_S f(x) = o_S)}{P(\mathcal{M}_1 f(x + \alpha) = o_1) \ldots P(\mathcal{M}_S f(x + \alpha) = o_S)}$$

$$\leq \prod_{s=1}^{S} \exp(\epsilon_s) = e^{(\sum_{s=1}^{S} \epsilon_s)}$$

*As a result, we have*

$$P\big(f(\mathcal{M}_1, \ldots, \mathcal{M}_S | x)\big) \leq e^{(\sum_i \epsilon_i)} P\big(f(\mathcal{M}_1, \ldots, \mathcal{M}_S | x + \alpha)\big)$$

*The sequential composition of the expected output is as:*

$$\mathbb{E}f(\mathcal{M}_1, \ldots, \mathcal{M}_S | x) = \int_0^1 P\big(f(\mathcal{M}_1, \ldots, \mathcal{M}_S | x) > t\big) dt$$

$$\leq e^{(\sum_s \epsilon_s)} \int_0^1 P\big(f(\mathcal{M}_1, \ldots, \mathcal{M}_S | x + \alpha) > t\big) dt$$

$$= e^{(\sum_s \epsilon_s)} \mathbb{E}f(\mathcal{M}_1, \ldots, \mathcal{M}_S | x + \alpha)$$

*Lemma 5 does hold.*

## J    PROOF OF THEOREM 5

**Proof 8** *$\forall \alpha \in l_p(1)$, from Lemma 5, with probability $\geq \eta$, we have that*

$$\hat{\mathbb{E}}f_k(\mathcal{M}_1, \ldots, \mathcal{M}_S | x + \alpha) \geq \frac{\hat{\mathbb{E}}f_k(\mathcal{M}_1, \ldots, \mathcal{M}_S | x)}{e^{(\sum_{s=1}^{s} \epsilon_s)}} \geq \frac{\hat{\mathbb{E}}_{lb}f_k(\mathcal{M}_1, \ldots, \mathcal{M}_S | x)}{e^{(\sum_{s=1}^{S} \epsilon_s)}} \tag{27}$$

*In addition, we also have*

$$\forall i \neq k : \hat{\mathbb{E}}f_{i:i\neq k}(\mathcal{M}_1, \ldots, \mathcal{M}_S | x + \alpha) \leq e^{(\sum_{s=1}^{S} \epsilon_s)} \hat{\mathbb{E}}f_{i:i\neq k}(\mathcal{M}_1, \ldots, \mathcal{M}_S | x)$$

$$\Rightarrow \forall i \neq k : \hat{\mathbb{E}}f_i(\mathcal{M}_1, \ldots, \mathcal{M}_S | x + \alpha) \leq e^{(\sum_{s=1}^{S} \epsilon_s)} \max_{i:i\neq k} \hat{\mathbb{E}}_{ub}f_i(\mathcal{M}_1, \ldots, \mathcal{M}_S | x) \tag{28}$$

*Using the hypothesis (Eq. 14) and the first inequality (Eq. 27), we have that*

$$\hat{\mathbb{E}}f_k(\mathcal{M}_1, \ldots, \mathcal{M}_S | x + \alpha) > \frac{e^{2(\sum_{s=1}^{S} \epsilon_s)} \max_{i:i\neq k} \hat{\mathbb{E}}_{ub}f_i(\mathcal{M}_1, \ldots, \mathcal{M}_S | x)}{e^{(\sum_{s=1}^{S} \epsilon_s)}}$$

$$> e^{(\sum_{s=1}^{S} \epsilon_s)} \max_{i:i\neq k} \hat{\mathbb{E}}_{ub}f_i(\mathcal{M}_1, \ldots, \mathcal{M}_S | x)$$

*Now, we apply the third inequality (Eq. 28), we have that*

$$\forall i \neq k : \hat{\mathbb{E}}f_k(\mathcal{M}_1, \ldots, \mathcal{M}_S | x + \alpha) > \hat{\mathbb{E}}f_i(\mathcal{M}_1, \ldots, \mathcal{M}_S | x + \alpha)$$

$$\Leftrightarrow \hat{\mathbb{E}}f_k(\mathcal{M}_1, \ldots, \mathcal{M}_S | x + \alpha) > \max_{i:i\neq k} \hat{\mathbb{E}}f_i(\mathcal{M}_1, \ldots, \mathcal{M}_S | x + \alpha)$$

*The Theorem 5 does hold.*

## K  PROOF OF PROPOSITION 1

**Proof 9** $\forall \alpha \in l_p(1)$, *by applying Theorem 5, we have*

$$\hat{\mathbb{E}}_{lb} f_k(\mathcal{M}_h, \mathcal{M}_x | x) > e^{2(\kappa \epsilon_r + \varphi \epsilon_r)} \max_{i:i \neq k} \hat{\mathbb{E}}_{ub} f_i(\mathcal{M}_h, \mathcal{M}_x | x)$$

$$> e^{2(\kappa + \varphi) \epsilon_r} \max_{i:i \neq k} \hat{\mathbb{E}}_{ub} f_i(\mathcal{M}_h, \mathcal{M}_x | x)$$

*Furthermore, by applying group privacy, we have that*

$$\forall \alpha \in l_p(\kappa + \varphi) : \hat{\mathbb{E}}_{lb} f_k(\mathcal{M}_h, \mathcal{M}_x | x) > e^{2\epsilon_r} \max_{i:i \neq k} \hat{\mathbb{E}}_{ub} f_i(\mathcal{M}_h, \mathcal{M}_x | x)$$

*By applying Proof 8, it is straight to have*

$$\forall \alpha \in l_p(\kappa + \varphi) : \hat{\mathbb{E}} f_k(\mathcal{M}_h, \mathcal{M}_x | x + \alpha) > \max_{i:i \neq k} \hat{\mathbb{E}} f_k(\mathcal{M}_h, \mathcal{M}_x | x + \alpha)$$

*with probability* $\geq \eta$. *Proposition 1 does hold.*

## L  EFFECTIVE MONTE CARLO ESTIMATION OF $\hat{\mathbb{E}} f(x)$

Recall that the Monte Carlo estimation is applied to estimate the expected value $\hat{\mathbb{E}} f(x) = \frac{1}{n} \sum_n f(x)_n$, where $n$ is the number of invocations of $f(x)$ with independent draws in the noise, i.e., $\frac{1}{m} Lap(0, \frac{\Delta_{\mathcal{R}}}{\epsilon_1})$ and $\frac{2}{m} Lap(0, \frac{\Delta_{\mathcal{R}}}{\epsilon_1})$ in our case. When $\epsilon_1$ is small (indicating a strong privacy protection), it causes a *notably large distribution shift between training and inference, given independent draws of the Laplace noise*.

In fact, let us denote a single draw in the noise as $\chi_1 = \frac{1}{m} Lap(0, \frac{\Delta_{\mathcal{R}}}{\epsilon_1})$ used to train the function $f(x)$, the model converges to the point that the noise $\chi_1$ and $2\chi_2$ need to be correspondingly added into $x$ and $h$ in order to make correct predictions. $\chi_1$ can be approximated as $Lap(\chi_1, \varrho)$, where $\varrho \to 0$. It is clear that independent draws of the noise $\frac{1}{m} Lap(0, \frac{\Delta_{\mathcal{R}}}{\epsilon_1})$ have *distribution shifts* with the fixed noise $\chi_1 \cong Lap(\chi_1, \varrho)$. These distribution shifts can also be large, when noise is large. We have experienced that these distribution shifts in having independent draws of noise to estimate $\hat{\mathbb{E}} f(x)$ can notably degrade the inference accuracy of the scoring function, when privacy budget $\epsilon_1$ is small resulting in a large amount of noise injected to provide strong privacy guarantees.

To address this, one solution is to increase the number of invocations of $f(x)$, i.e., $n$, to a huge number per prediction. However, this is impractical in real-world scenarios. We propose a novel way to draw independent noise following the distribution of $\chi_1 + \frac{1}{m} Lap(0, \frac{\Delta_{\mathcal{R}}}{\epsilon_1}/\psi)$ for the input $x$ and $2\chi_2 + \frac{2}{m} Lap(0, \frac{\Delta_{\mathcal{R}}}{\epsilon_1}/\psi)$ for the affine transformation $h$, where $\psi$ is a hyper-parameter to control the distribution shifts. This approach works well and does not affect the DP bounds and the provable robustness condition, since: **(1)** Our mechanism achieves both DP and provable robustness in the training process; and **(2)** It is clear that $\hat{\mathbb{E}} f(x) = \frac{1}{n} \sum_n f(x)_n = \frac{1}{n} \sum_n g\big(a(x + \chi_1 + \frac{1}{m} Lap_n(0, \frac{\Delta_{\mathcal{R}}}{\epsilon_1}/\psi), \theta_1) + 2\chi_2 + \frac{2}{m} Lap_n(0, \frac{\Delta_{\mathcal{R}}}{\epsilon_1}/\psi), \theta_2\big)$, where $Lap_n(0, \frac{\Delta_{\mathcal{R}}}{\epsilon_1}/\psi)$ is the $n$-th draw of the noise. When $n \to \infty$, $\hat{\mathbb{E}} f(x)$ will converge to $\frac{1}{n} \sum_n g\big(a(x + \chi_1, \theta_1) + 2\chi_2, \theta_2\big)$, which aligns well with the convergence point of the scoring function $f(x)$. Injecting $\chi_1$ and $2\chi_2$ to $x$ and $h$ during the estimation of $\hat{\mathbb{E}} f(x)$ yields better performance, without affecting the DP and the robustness bounds.

## M  MODEL CONFIGURATIONS

The MNIST database consists of handwritten digits (Lecun et al., 1998). Each example is a $28 \times 28$ size gray-level image. The CIFAR-10 dataset consists of color images belonging to 10 classes, i.e., airplanes, dogs, etc. The dataset is split into 50,000 training samples and 10,000 test samples (Krizhevsky & Hinton, 2009). The experiments were conducted on a single GPU, i.e., NVIDIA GTX TITAN X, 12 GB with 3,072 CUDA cores. All the models share the same structure, consisting of 2 and 3 convolutional layers, respectively for MNIST and CIFAR-10 datasets.

Both fully-connected and convolution layers can be applied in the representation learning model $a(x, \theta_1)$. Given convolution layer, the computation of each feature map needs to be DP; since each of them independently reads a local region of input neurons. Therefore, the sensitivity $\Delta_{\mathcal{R}}$ can be considered the maximal sensitivity given any single feature map in the first affine transformation layer. In addition, each hidden neuron can only be used to reconstruct a unit patch of input units. That results in $d$ (Lemma 2) being the size of the unit patch connected to each hidden neuron, e.g., $d = 9$ given a $3 \times 3$ unit patch, and $\beta$ is the number of hidden neurons in a feature map.

*MNIST:* We used two convolutional layers (32 and 64 features). Each hidden neuron connects with a 5x5 unit patch. A fully-connected layer has 256 units. The batch size $m$ was set to 2,499, $\xi = 1$, $\psi = 2$. I-FGSM, MIM, and MadryEtAl were used to draft $l_\infty(\mu)$ adversarial examples in training, with $T_\mu = 10$. Learning rate $\varrho_t$ was set to $1e - 4$. Given a predefined total privacy budget $\epsilon_t$, $\epsilon_2$ is set to be 0.1, and $\epsilon_1$ is computed as: $\epsilon_1 = \frac{\epsilon_t - \epsilon_2}{(1 + 1/\gamma + 1/\gamma_{\mathbf{x}})}$. This will guarantee that $(\epsilon_1 + \epsilon_1/\gamma_{\mathbf{x}} + \epsilon_1/\gamma + \epsilon_2) = \epsilon_t$. $\Delta_{\mathcal{R}} = (14^2 + 2) \times 25$ and $\Delta_{\mathcal{L}2} = 2 \times 256$.

*CIFAR-10:* We used three convolutional layers (128, 128, and 256 features). Each hidden neuron connects with a 4x4 unit patch in the first layer, and a 5x5 unit patch in other layers. One fully-connected layer has 256 neurons. The batch size $m$ was set to 1,851, $\xi = 1.5$, $\psi = 10$, and $T_\mu = 3$. The ensemble of attacks $A$ includes I-FGSM, MIM, and MadryEtAl. We use data augmentation, including random crop, random flip, and random contrast. Learning rate $\varrho_t$ was set to $5e - 2$. In the CIFAR-10 dataset, $\epsilon_2$ is set to $(1 + r/3.0)$ and $\epsilon_1 = (1 + 2r/3.0)/(1 + 1/\gamma + 1/\gamma_{\mathbf{x}})$, where $r \geq 0$ is a ratio to control the total privacy budget $\epsilon_t$ in our experiment. For instance, given $r = 0$, we have that $\epsilon_t = (\epsilon_1 + \epsilon_1/\gamma_{\mathbf{x}} + \epsilon_1/\gamma + \epsilon_2) = 2$. $\Delta_{\mathcal{R}} = 3 \times (14^2 + 2) \times 16$ and $\Delta_{\mathcal{L}2} = 2 \times 256$.

**Computational Efficiency and Scalability.** In terms of computation efficiency, our mechanism does not consume any extra computational resources to train the model, compared with existing DP-preserving algorithms in deep learning (Phan et al., 2016; 2017b;a). The model invocations to approximate the robustness bounds can further be efficiently performed in a parallel process. Regarding the scalability, with remarkably tightened global sensitivities, the impact of the size of deep neural networks in terms of the number of hidden layers and hidden neurons is significantly remedied, since 1) $\Delta_{\mathcal{R}}$ and $\Delta_{\mathcal{L}2}$ are small, 2) we do not need to inject any noise into the computation of the network $g(\cdot)$, and 3) we do not redraw the noise in each training step $t$. In addition, our mechanism is not restricted to the type of activation functions. That is similar to (Lecuyer et al., 2018; Phan et al., 2019). As a result, our mechanism has a great potential to be applied in larger deep neural networks using larger datasets. Extensively investigating this property requires further study from both research and practice communities.

# N  APPROXIMATION ERROR BOUNDS

To compute how much error our polynomial approximation approaches (i.e., truncated Taylor expansions), $\widetilde{\mathcal{R}}_{B_t}(\theta_1)$ (Eq. 6) and $\mathcal{L}_{\overline{B}_t}(\theta_2)$, incur, we directly apply Lemma 4 in (Phan et al., 2016), Lemma 3 in (Zhang et al., 2012), and the well-known error bound results in (Apostol, 1967). Note that $\widetilde{\mathcal{R}}_{B_t}(\theta_1)$ is the 1st-order Taylor series and $\mathcal{L}_{\overline{B}_t}(\theta_2)$ is the 2nd-order Taylor series. Let us closely follow (Phan et al., 2016; Zhang et al., 2012; Apostol, 1967) to adapt their results into our scenario, as follows:

Given the truncated function $\widetilde{\mathcal{R}}_{B_t}(\theta_1) = \sum_{x_i \in B_t} \sum_{j=1}^d \sum_{l=1}^2 \sum_{r=0}^1 \frac{\mathbf{F}_{lj}^{(r)}(0)}{r!} (\theta_{1j} h_i)^r$, the original Taylor polynomial function $\widehat{\mathcal{R}}_{B_t}(\theta_1) = \sum_{x_i \in B_t} \sum_{j=1}^d \sum_{l=1}^\infty \sum_{r=0}^1 \frac{\mathbf{F}_{lj}^{(r)}(0)}{r!} (\theta_{1j} h_i)^r$, the average error of the approximation is bounded as

$$\frac{1}{|B_t|} |\widehat{\mathcal{R}}_{B_t}(\widetilde{\theta}_1) - \widehat{\mathcal{R}}_{B_t}(\widehat{\theta}_1)| \leq \frac{4e \times d}{(1 + e)^2} \tag{29}$$

$$\frac{1}{|B_t|} |\widehat{\mathcal{L}}_{B_t}(\widetilde{\theta}_2) - \widehat{\mathcal{L}}_{B_t}(\widehat{\theta}_2)| \leq \frac{e^2 + 2e - 1}{e(1 + e)^2} \times K \tag{30}$$

where $\widehat{\theta}_1 = \arg\min_{\theta_1} \widehat{\mathcal{R}}_{B_t}(\theta_1)$, $\widetilde{\theta}_1 = \arg\min_{\theta_1} \widetilde{\mathcal{R}}_{B_t}(\theta_1)$, $\widehat{\mathcal{L}}_{B_t}(\theta_2)$ is the original Taylor polynomial function of $\sum_{x_i \in B_t} \mathcal{L}(f(\overline{x}_i, \theta_2), y_i)$, $\widehat{\theta}_2 = \arg\min_{\theta_2} \widehat{\mathcal{L}}_{B_t}(\theta_2)$, $\widetilde{\theta}_2 = \arg\min_{\theta_2} \mathcal{L}_{B_t}(\theta_2)$.

**Proof 10** *Let $U = \max_{\theta_1} \left( \widehat{\mathcal{R}}_{B_t}(\theta_1) - \widetilde{\mathcal{R}}_{B_t}(\theta_1) \right)$ and $S = \min_{\theta_1} \left( \widehat{\mathcal{R}}_{B_t}(\theta_1) - \widetilde{\mathcal{R}}_{B_t}(\theta_1) \right)$.*

*We have that $U \geq \widehat{\mathcal{R}}_{B_t}(\widetilde{\theta}_1) - \widetilde{\mathcal{R}}_{B_t}(\widetilde{\theta}_1)$ and $\forall \theta_1^* : S \leq \widehat{\mathcal{R}}_{B_t}(\theta_1^*) - \widetilde{\mathcal{R}}_{B_t}(\theta_1^*)$. Therefore, we have*

$$\widehat{\mathcal{R}}_{B_t}(\widetilde{\theta}_1) - \widetilde{\mathcal{R}}_{B_t}(\widetilde{\theta}_1) - \widehat{\mathcal{R}}_{B_t}(\theta_1^*) + \widetilde{\mathcal{R}}_{B_t}(\theta_1^*) \leq U - S \tag{31}$$

$$\Leftrightarrow \widehat{\mathcal{R}}_{B_t}(\widetilde{\theta}_1) - \widehat{\mathcal{R}}_{B_t}(\theta_1^*) \leq U - S + \left( \widetilde{\mathcal{R}}_{B_t}(\widetilde{\theta}_1) - \widetilde{\mathcal{R}}_{B_t}(\theta_1^*) \right) \tag{32}$$

*In addition, $\widetilde{\mathcal{R}}_{B_t}(\widetilde{\theta}_1) - \widetilde{\mathcal{R}}_{B_t}(\theta_1^*) \leq 0$, it is straightforward to have:*

$$\widehat{\mathcal{R}}_{B_t}(\widetilde{\theta}_1) - \widehat{\mathcal{R}}_{B_t}(\theta_1^*) \leq U - S \tag{33}$$

*If $U \geq 0$ and $S \leq 0$ then we have:*

$$|\widehat{\mathcal{R}}_{B_t}(\widetilde{\theta}_1) - \widehat{\mathcal{R}}_{B_t}(\theta_1^*)| \leq U - S \tag{34}$$

*Eq. 34 holds for every $\theta_1^*$, including $\widehat{\theta}_1$. Eq. 34 shows that the error incurred by truncating the Taylor series approximate function depends on the maximum and minimum values of $\widehat{\mathcal{R}}_{B_t}(\theta_1) - \widetilde{\mathcal{R}}_{B_t}(\theta_1)$. This is consistent with (Phan et al., 2016; Zhang et al., 2012). To quantify the magnitude of the error, we rewrite $\widehat{\mathcal{R}}_{B_t}(\theta_1) - \widetilde{\mathcal{R}}_{B_t}(\theta_1)$ as:*

$$\widehat{\mathcal{R}}_{B_t}(\theta_1) - \widetilde{\mathcal{R}}_{B_t}(\theta_1) = \sum_{j=1}^{d} \left( \widehat{\mathcal{R}}_{B_t}(\theta_{1j}) - \widetilde{\mathcal{R}}_{B_t}(\theta_{1j}) \right) \tag{35}$$

$$= \sum_{j=1}^{d} \left( \sum_{i=1}^{|B_t|} \sum_{l=1}^{2} \sum_{r=3}^{\infty} \frac{\mathbf{F}_{lj}^{(r)}(z_{lj})}{r!} \left( g_{lj}(x_i, \theta_{1j}) - z_{lj} \right)^r \right) \tag{36}$$

*where $g_{1j}(x_i, \theta_{1j}) = \theta_{1j} h_i$ and $g_{2j}(x_i, \theta_{1j}) = \theta_{1j} h_i$.*

*By looking into the remainder of Taylor expansion for each $j$ (i.e., following (Phan et al., 2016; Apostol, 1967)), with $z_j \in [z_{lj} - 1, z_{lj} + 1]$, $\frac{1}{|B_t|} \left( \widehat{\mathcal{R}}_{B_t}(\theta_{1j}) - \widetilde{\mathcal{R}}_{B_t}(\theta_{1j}) \right)$ must be in the interval $\left[ \sum_l \frac{\min_{z_j} \mathbf{F}_{lj}^{(2)}(z_j)(z_j - z_{lj})^2}{2!}, \sum_l \frac{\max_{z_j} \mathbf{F}_{lj}^{(2)}(z_j)(z_j - z_{lj})^2}{2!} \right]$. If $\sum_l \frac{\max_{z_j} \mathbf{F}_{lj}^{(2)}(z_j)(z_j - z_{lj})^2}{2!} \geq 0$ and $\sum_l \frac{\min_{z_j} \mathbf{F}_{lj}^{(2)}(z_j)(z_j - z_{lj})^2}{2!} \leq 0$, then we have that $|\frac{1}{|B_t|} \left( \widehat{\mathcal{R}}_{B_t}(\theta_1) - \widetilde{\mathcal{R}}_{B_t}(\theta_1) \right)| \leq \sum_{j=1}^{d} \sum_l \frac{\max_{z_j} \mathbf{F}_{lj}^{(2)}(z_j)(z_j - z_{lj})^2 - \min_{z_j} \mathbf{F}_{lj}^{(2)}(z_j)(z_j - z_{lj})^2}{2!}$. This can be applied to the case of our auto-encoder, as follows:*

*For the functions $\mathbf{F}_{1j}(z_j) = x_{ij} \log(1 + e^{-z_j})$ and $\mathbf{F}_{2j}(z_j) = (1 - x_{ij}) \log(1 + e^{z_j})$, we have $\mathbf{F}_{1j}^{(2)}(z_j) = \frac{x_{ij} e^{-z_j}}{(1 + e^{-z_j})^2}$ and $\mathbf{F}_{2j}^{(2)}(z_j) = (1 - x_{ij}) \frac{e^{z_j}}{(1 + e^{z_j})^2}$. It can be verified that $\arg\min_{z_j} \mathbf{F}_{1j}^{(2)}(z_j) = \frac{-e}{(1+e)^2} < 0$, $\arg\max_{z_j} \mathbf{F}_{1j}^{(2)}(z_j) = \frac{e}{(1+e)^2} > 0$, $\arg\min_{z_j} \mathbf{F}_{2j}^{(2)}(z_j) = 0$, and $\arg\max_{z_j} \mathbf{F}_{2j}^{(2)}(z_j) = \frac{2e}{(1+e)^2} > 0$. Therefore, the average error of the approximation is at most:*

$$\frac{1}{|B_t|} |\widehat{\mathcal{R}}_{B_t}(\widetilde{\theta}_1) - \widehat{\mathcal{R}}_{B_t}(\widehat{\theta}_1)| \leq \left[ \left( \frac{e}{(1+e)^2} - \frac{-e}{(1+e)^2} \right) + \frac{2e}{(1+e)^2} \right] \times d = \frac{4e \times d}{(1+e)^2} \tag{37}$$

*Consequently, Eq. 29 does hold. Similarly, by looking into the remainder of Taylor expansion for each label $k$, Eq. 30 can be proved straightforwardly. In fact, by using the 2nd-order Taylor series with $K$ categories, we have that: $\frac{1}{|B_t|} |\widehat{\mathcal{L}}_{B_t}(\widetilde{\theta}_2) - \widehat{\mathcal{L}}_{B_t}(\widehat{\theta}_2)| \leq \frac{e^2 + 2e - 1}{e(1+e)^2} \times K$.*

## O  COMPLETE EXPERIMENTAL RESULTS

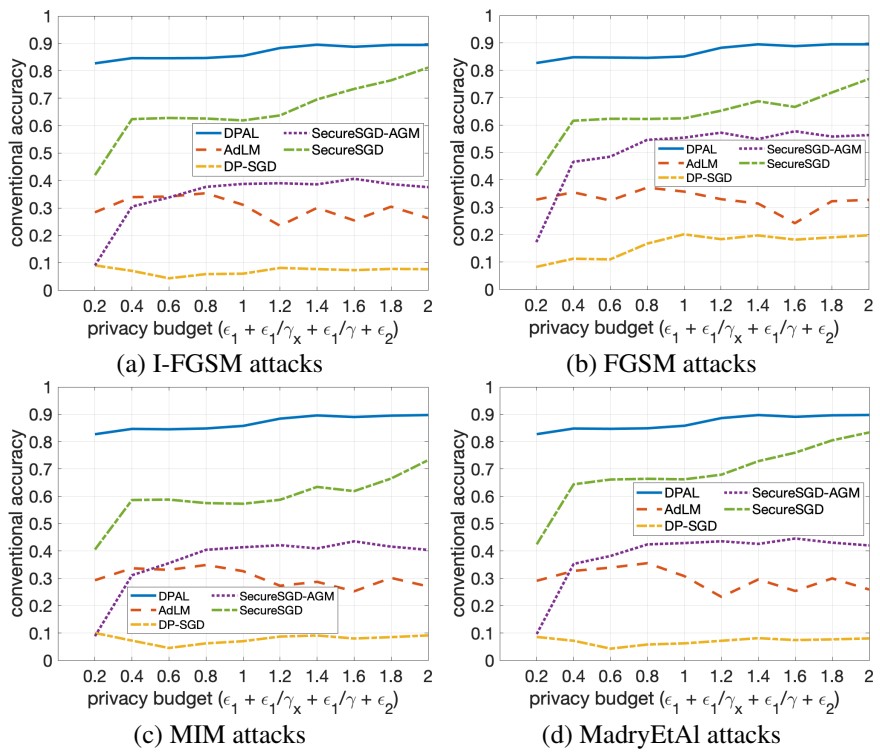

Figure 2: Conventional accuracy on the MNIST dataset given the privacy budget, under $l_\infty(\mu_a = 0.2)$.

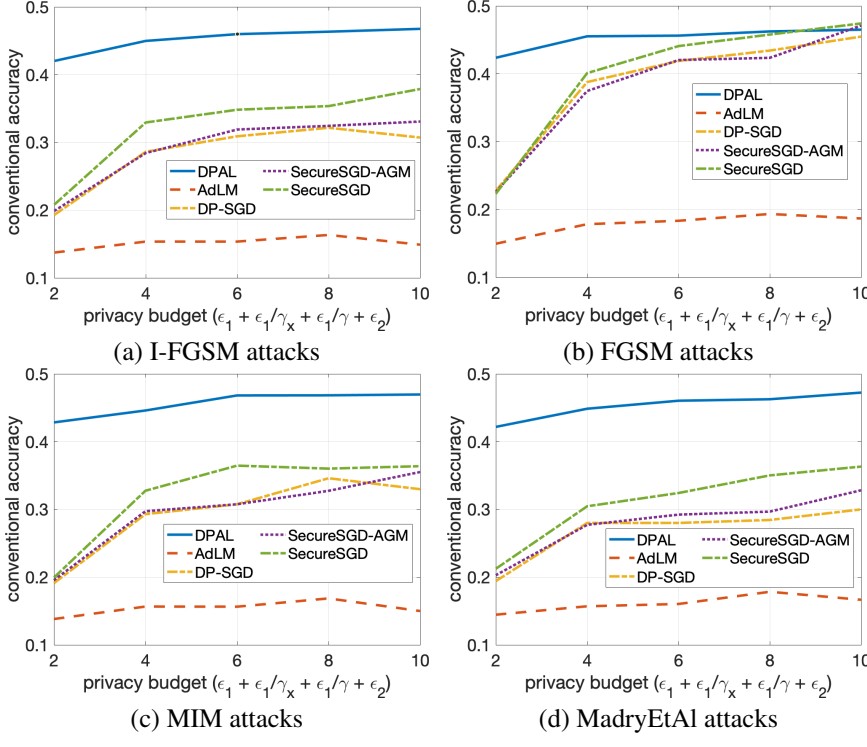

Figure 3: Conventional accuracy on the CIFAR-10 dataset given the privacy budget, under $l_\infty(\mu_a = 0.2)$.

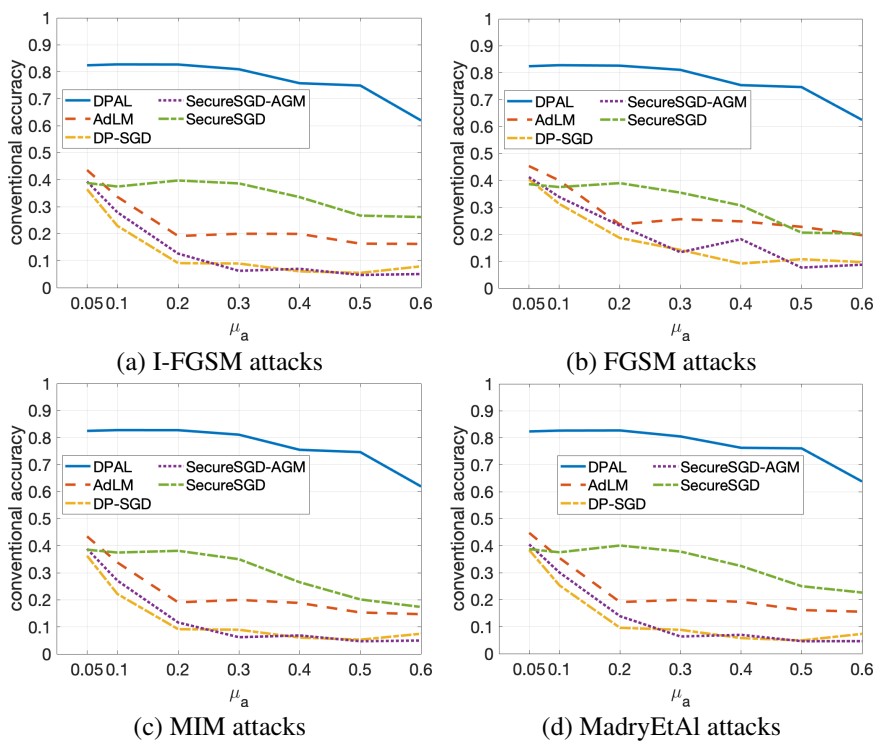

(a) I-FGSM attacks

(b) FGSM attacks

(c) MIM attacks

(d) MadryEtAl attacks

Figure 4: Conventional accuracy on the MNIST dataset given the attack size $\mu_a$, under $(\epsilon_1 + \epsilon_1/\gamma_{\mathbf{x}} + \epsilon_1/\gamma + \epsilon_2) = 0.2$ (tight DP protection).

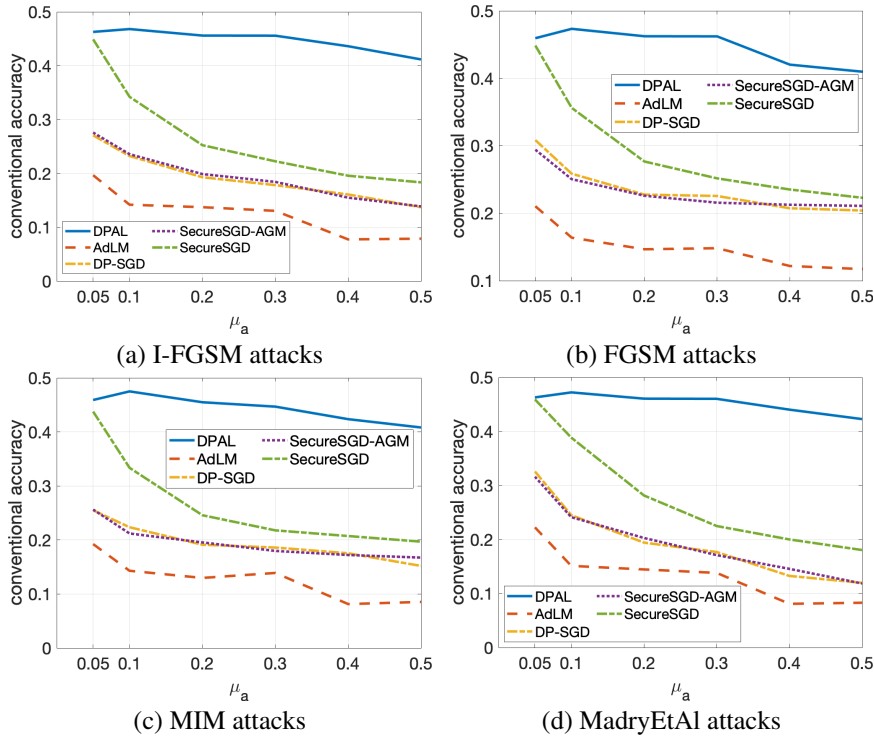

(a) I-FGSM attacks

(b) FGSM attacks

(c) MIM attacks

(d) MadryEtAl attacks

Figure 5: Conventional accuracy on the CIFAR-10 dataset given the attack size $\mu_a$, under $(\epsilon_1 + \epsilon_1/\gamma_{\mathbf{x}} + \epsilon_1/\gamma + \epsilon_2) = 2$ (tight DP protection).

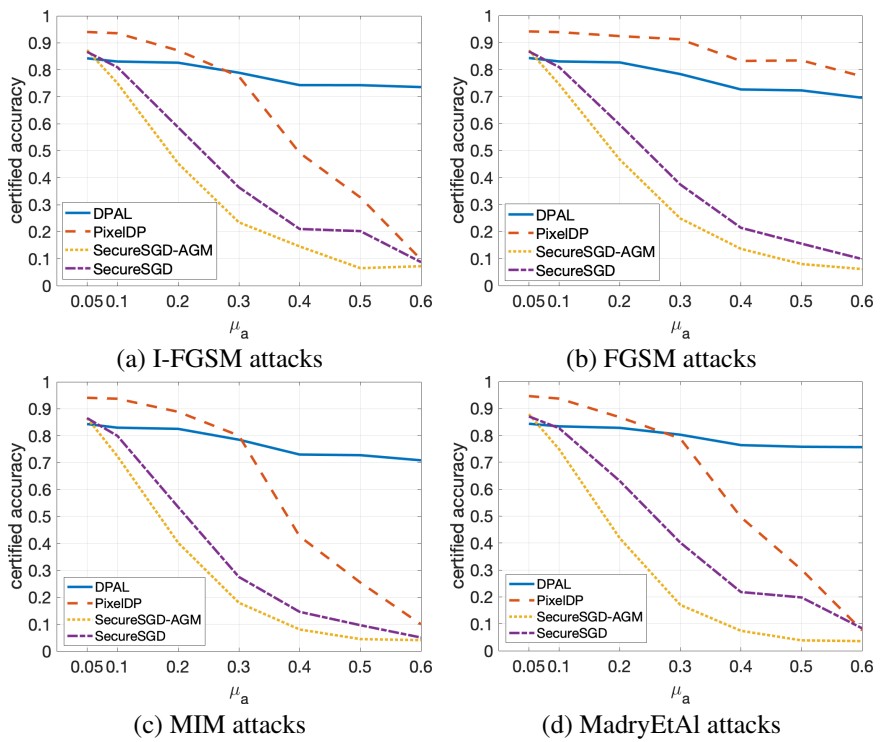

(a) I-FGSM attacks        (b) FGSM attacks

(c) MIM attacks        (d) MadryEtAl attacks

Figure 6: Certified accuracy on the MNIST dataset. The privacy budget is set to 1.0 (tight DP protection).

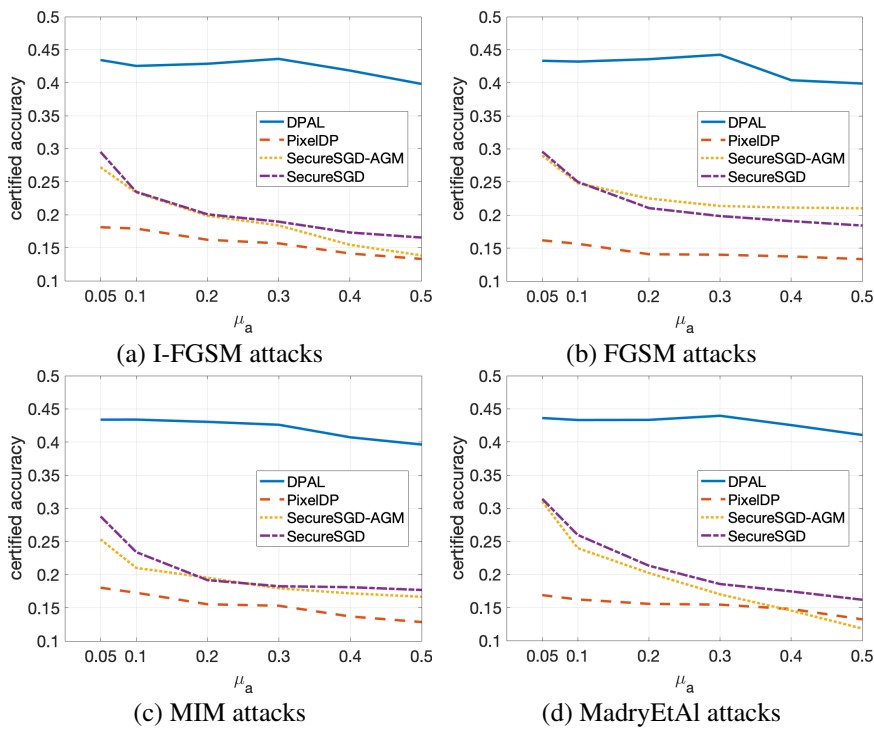

(a) I-FGSM attacks        (b) FGSM attacks

(c) MIM attacks        (d) MadryEtAl attacks

Figure 7: Certified accuracy on the CIFAR-10 dataset. The privacy budget is set to 2 (tight DP protection).

