# OpenReview forum: "Differential Privacy in Adversarial Learning with Provable Robustness"
_ICLR.cc/2020/Conference — Reject_

### Official Review · AnonReviewer3 · 2019-10-19
**Official Blind Review #3**

**Rating:** 3

**Review:**

This paper focus on providing both differential privacy and adversarial robustness to machine learning models. The authors propose an algorithm called differentially private adversarial learning (DPAL) to achieve such goal. DPAL consists two sub-models: (1) An auto-encoder to extract feature representation; and (2) A classifier takes the embedding of encoder and return the predict logits. The auto encoder uses the reconstruction loss and the classifier uses the similar loss as in adversarial learning.

The auto-encoder takes both real and adversarial examples as input. In order to guarantee  differential privacy to both sub-models, the authors use Functional Mechanism (Zhang et al., 2012), which perturbs the objective function to guarantee the objective is differentially private. To apply functional mechanism, the authors use the 1st-order polynomial approximation of both objective functions (by Taylor Expansion). Laplace noise is injected to both network input and the output of encoder to ensure the objective functions are differentially private (i.e. for any given weight, the influence of individual record on loss value is DP preserved). The authors further show the noise to guarantee privacy can be convert to PixelDP (Lecuyer et al., 2018) and therefore leads to provable robustness against adversarial examples.

Overall, this paper studies an interesting setting when privacy guarantee and adversarial robustness are both needed in machine learning model and proposes an algorithm to achieve such goal. The paper is well-organized. However, the paper suffers from several critical questions. I do not support the acceptance unless the following questions are well addressed.

1.	The approximation is necessary to derive the privacy guarantee for the objective function. But the paper does not provide either theoretical analysis on the approximation error or empirical result of the distance between the approximate and real function values.  Another way is to use the approximate loss as the objective function to train the models and see how the model performance changes.
2. The differentially private objective does not lead to differentially private parameters.  In the proof of your Theorem 1, it is clear that the reconstruction loss in Algorithm 1 is differentially private. However, this does not lead to differentially private parameters. In Zhang et al., 2012, the resulting parameters are differentially private only if finding the minimizer of the perturbed objective does not involve any additional information from the original database (see the proof of their Theorem 1). One way is using data-independent grid search to find parameters with low loss. However, Algorithm 1 uses gradient descent to update the parameters, which needs to access the original database at each update. Though the paper uses a perturbed database, the perturbation is not sufficient to provide the privacy level as claimed in Theorem 1.

Minor comments:
1.	When is \chi_2 in Algorithm 1 being used? In Algorithm1, I do not see its appearance apart from its definition.
2.	At the end of Introduction, you claim DPAL establishes the first connection between DP preservation and provable robustness. However, in the experiments section, SecureSGD and SecureSGD-AGM are given as baseline algorithms which are DP-preserving algorithms with provable robustness.


**Experience Assessment:**

I have published one or two papers in this area.

**Review Assessment: Checking Correctness Of Derivations And Theory:**

I assessed the sensibility of the derivations and theory.

**Review Assessment: Checking Correctness Of Experiments:**

I assessed the sensibility of the experiments.

**Review Assessment: Thoroughness In Paper Reading:**

I read the paper at least twice and used my best judgement in assessing the paper.

---

> ### Author Response · Authors · 2019-11-09
> **Response to Review 3**
>
> We thank all reviewers for the constructive feedback. We have thoroughly addressed all the questions raised in our following responses as follows.
>
> Q1: The approximation error bounds are needed.
>
> A: The approximation errors bounds have been derived and added to the Appendix N. To compute how much error our polynomial approximation approaches (i.e., truncated Taylor expansions), $\widetilde{\mathcal{R}}_{B_t}(\theta_1)$ and $\mathcal{L}_{\overline{B}_t}\big(\theta_2\big)$, incur, we directly apply Lemma 4 in (Phan et al., 2016), Lemma 3 in (Zhang et al., 2012), and the well-known remainder error bounds in (Apostol, 1967). Note that $\widetilde{\mathcal{R}}_{B_t}(\theta_1)$ is the 1st-order Taylor series and $\mathcal{L}_{\overline{B}_t}\big(\theta_2\big)$ is the 2nd-order Taylor series. Let us closely follow (Phan et al., 2016; Zhang et al., 2012; Apostol, 1967) to adapt their results into our scenario, as follows:
>
> Given the truncated function $\widetilde{\mathcal{R}}_{B_t}(\theta_1) = \sum_{x_i \in B_t} \sum_{j = 1}^d \sum_{l=1}^{2} \sum_{r = 0}^{1} \frac{\mathbf{F}^{(r)}_{lj}(0)}{r!}\big(\theta_{1j}h_i\big)^r$, the original Taylor polynomial function $\widehat{\mathcal{R}}_{B_t}(\theta_1) = \sum_{x_i \in B_t} \sum_{j = 1}^d \sum_{l=1}^{\infty} \sum_{r = 0}^{1} \frac{\mathbf{F}^{(r)}_{lj}(0)}{r!}\big(\theta_{1j}h_i\big)^r$, the average error of the approximation is bounded as
> $$\frac{1}{|B_t|}|\widehat{\mathcal{R}}_{B_t}(\widetilde{\theta}_1) - \widehat{\mathcal{R}}_{B_t}(\widehat{\theta}_1)| \leq \frac{4e \times d}{(1+e)^2}$$,
> $$\frac{1}{|B_t|}|\widehat{\mathcal{L}}_{B_t}(\widetilde{\theta}_2) - \widehat{\mathcal{L}}_{B_t}(\widehat{\theta}_2)| \leq \frac{e^2 + 2e - 1}{e(1+e)^2} \times K$$
> where $\widehat{\theta}_1 = \arg \min_{\theta_1} \widehat{\mathcal{R}}_{B_t}(\theta_1)$, $\widetilde{\theta}_1 = \arg \min_{\theta_1} \widetilde{\mathcal{R}}_{B_t}(\theta_1)$, $\widehat{\mathcal{L}}_{B_t}(\theta_2)$ is the original Taylor polynomial function of $\sum_{x_i \in B_t} \mathcal{L}\big(f(\overline{x}_i, \theta_2), y_i\big)$,
> $\widehat{\theta}_2 = \arg \min_{\theta_2} \widehat{\mathcal{L}}_{B_t}(\theta_2)$, $\widetilde{\theta}_2 = \arg \min_{\theta_2} {\mathcal{L}}_{B_t}(\theta_2)$. The proof is in Appendix N.
>
> The reviewer suggested to "use the approximate loss as the objective function to train the models and see how the model performance changes." This is an interesting suggestion. However, this could result in violating DP guarantees; thus it may only be applied with great care. We will consider this suggestion in our feature work.
>
> Q2: The paper uses a perturbed database, the perturbation is not sufficient to provide the privacy level as claimed in Theorem 1.
>
> A: To clearly clarify this question, we have reordered the Lemma 3 to be presented before Theorem 1. This will allow us to revise the Theorem 1 in order to clearly prove that the Alg. 1 does achieve $(\epsilon_1/\gamma_\mathbf{x} + \epsilon_1)$-DP in learning $\theta_1$. In fact, the parameter optimization of $\overline{\mathcal{R}}_{\overline{B}_t}(\theta_1)$ only uses the perturbed data $\overline{B}_t$, which is $(\epsilon_1/\gamma_{\mathbf{x}})$-DP (Lemma 3), in the computations of $h_i$, $\overline{h}_i$, $\widetilde{x}_i$, parameter gradients, and gradient descents at each step. These operations do not access the original dataset $B_t$; therefore, they do not incur any additional information from the original data (the post-processing property in DP (Dwork and Roth, 2014)). As a result, the total privacy budget to learn the perturbed optimal parameters $\overline{\theta}_1$ in Alg. 1 is $(\epsilon_1/\gamma_{\mathbf{x}} + \epsilon_1)$-DP, where the privacy budget $\epsilon_1/\gamma_{\mathbf{x}}$ is used to perturb the original data $B_t$. Applying gradient descent does not incur any additional information from the original data at each step.
>
> This result clearly does not affect our end-to-end privacy analysis in Theorem 4, as well as it does not affect our experimental results. The revised Theorem 1 has been added to our revision for a better presentation. In addition, Theorem 1 in (Zhang et al., 2012) may not be directly applied in our scenario, since (Zhang et al., 2012) does not use the $(\epsilon_1/\gamma_{\mathbf{x}})$-DP perturbed data as in our mechanism.
>
> Q3: When is $\chi_2$ in Algorithm 1 is used?
>
> A: $\chi_2$ has been added to the Alg. 1 as it is injected to the first hidden layer.
>
> Q4: You claim DPAL establishes the first connection between DP preservation and provable robustness.
>
> A: The reviewer did miss the term "in adversarial learning" in our statement. In our original version, our statement mentioned that: "To our knowledge, our mechanism establishes the first connection between DP preservation and provable robustness against adversarial examples in adversarial learning." SecureSGD and SecureSGD-AGM are not adversarial training algorithms (Phan et al., 2019).

---

> ### Author Response · Authors · 2019-11-09
> **Response to Review 3 - Reference**
>
> [1] T. Apostol.Calculus. John Wiley & Sons, 1967.
> [2] Cynthia Dwork and Aaron Roth. The algorithmic foundations of differential privacy. Found. TrendsTheor. Comput. Sci.,  9(3&#8211;4):211–407, August 2014.
> [3] Mathias Lecuyer, Vaggelis Atlidakis, Roxana Geambasu, Daniel Hsu, and Suman Jana.  Certified robustness to adversarial examples with differential privacy. In S&P, 2019.
> [4] NhatHai Phan, Yue Wang, Xintao Wu, and Dejing Dou. Differential privacy preservation for deep auto-encoders: an application of human behavior prediction. InAAAI’16, pp. 1309–1316, 2016.
> [5] NhatHai Phan, Minh N. Vu, Yang Liu, Ruoming Jin, Dejing Dou, Xintao Wu, and My T. Thai. Heterogeneous gaussian mechanism: Preserving differential privacy in deep learning with provable robustness. In IJCAI, 2019.
> [6] Jun Zhang, Zhenjie Zhang, Xiaokui Xiao, Yin Yang, and Marianne Winslett. Functional mechanism: regression analysis under differential privacy.PVLDB, 5(11):1364–1375, 2012.

---

### Official Review · AnonReviewer1 · 2019-10-21
**Official Blind Review #1**

**Rating:** 3

**Review:**

This paper propose an algorithm with DP preservation to train adversarially robust neural networks. To preserve DP, a single-layer linear autoencoder with shared weights is learned to extract features from training data, whose encoder is used to extract private features for the training and inference of a deeper network. To enhance robustness against various attacks, adversarial examples crafted with such attacks are injected into the training set in this algorithm. Guarantees of privacy preservation for training on both clean data and adversarial examples for the autoencoder and the inference network are given. Certified robustness of the smoothed classifier is also given, which depends on the privacy budget of each compositing mechanism. Experimental evaluations of two small (2 and 3 conv layers) networks are given on the MNIST and CIFAR10 dataset, showing improved conventional accuracy on clean samples and adversarial attacks, and certified accuracy than 4 baseline privacy-preserving algorithms.

To my knowledge, this paper provides the most comprehensive analysis so far about privacy preservation in the process of enhancing empirical adversarial robustness, as well as the impact of privacy preservation on the certified robustness of the smoothed classifier. However, the current version is quite difficult to follow for people without DP background, with some settings even conflict with other papers on adversarial robustness, and I do have some doubts about the experimental results. Specifically,

1. What is the role of adversarial examples in the proposed algorithm? Perhaps I missed something, but the authors have not shown how it is related to certified robustness in the paper. The adversarial examples seem to be used only for improving the empirical accuracy against various adversarial attacks. Empirical adversarial robustness is usually down weighted to me when certified robustness is given, therefore an algorithm enhancing only the empirical one does not seem so interesting.

2. What is even more against my intuition at first glance is that the empirical adversarial accuracy in Figure 4 is lower than the certified accuracy in Figure 6 in some situations. After a while I realized that the conventional accuracy against attacks in Figure 4,5 and the certified accuracy in Figure 6,7 are actually talking about different models, where the first one is the deterministic (or one-random-sample) inference network but the second one is for the smoothed classifier (as referred to in by Cohen et al. 2019, or the expectation of the inference network). This distinction should be addressed, since only one model can be chosen at deployment.

3. The attacks used for evaluating the empirical adversarial robustness are too weak. Only 10 steps are used. More iterative steps, e.g., 1000 or 10000 step PGD I-FGSM, should be provided to reveal the actual robustness of the networks in Figure 4,5.

4. The results of the proposed algorithm in Figure 7 is much better than the state-of-the-art, but I cannot see clearly from this paper how such improvement is achieved. With a ResNet110, [1] and [2] can only achieve around 30% certified accuracy when the maximum l_infty norm is 8/255. However, in Figure 7, the proposed algorithm is able to keep the certified accuracy above 40% at much larger perturbations, with only a 3-conv-layer network. If I am understanding the numbers correctly, could the authors explain clearly where such improvements come from?

Therefore, I tend to reject the paper before my concerns are addressed.

[1] "Filling the Soap Bubbles: Efficient Black-Box Adversarial Certification with Non-Gaussian Smoothing", https://openreview.net/forum?id=Skg8gJBFvr
[2] Salman, Hadi, et al. "Provably Robust Deep Learning via Adversarially Trained Smoothed Classifiers." NeurIPS (2019).

**Experience Assessment:**

I have read many papers in this area.

**Review Assessment: Checking Correctness Of Derivations And Theory:**

I assessed the sensibility of the derivations and theory.

**Review Assessment: Checking Correctness Of Experiments:**

I assessed the sensibility of the experiments.

**Review Assessment: Thoroughness In Paper Reading:**

I made a quick assessment of this paper.

---

> ### Author Response · Authors · 2019-11-09
> **Response to Review 1 - Part 1**
>
> We thank all reviewers for the constructive feedback. We have thoroughly addressed all the questions raised in our following responses as follows.
>
> Q1: What is the role of adversarial examples in the proposed algorithm?
>
> A: Our focus is to establish the first connection between DP preservation to protect the training data, typical adversarial learning using adversarial examples, and DP-certified robustness bounds. Adversarial examples are used to practically strengthen our DP-based certified robustness bounds. Other robustness analysis [1, 2] given adversarial examples is potentially applicable on top of our mechanism. However, there is an unknown correlation between the robustness bounds in [1, 2] and DP-based robustness bounds and DP guarantees of the training data in our paper, since: (1) The magnitude of noise used in their smoothing function could affect the DP bounds and DP protections; (2) There is an additional difference between our mechanism and [1, 2] contributing to this unknown correlation, that is noise is injected into both input and the latent space (the first hidden layer) to derive our DP-based composition robustness bounds. Meanwhile, noise only is injected into the input $x$ in [1, 2]; and (3) Our training mechanism is significantly different from the training algorithms in [1, 2], in order to preserve DP in learning model parameters.
>
> Understanding this unexplored connection is non-trivial. This remains a largely open question, requiring significant efforts to answer. Note that pure robust models can incur privacy risk [3].
>
> Q2: This distinction should be addressed, since only one model can be chosen at deployment.
>
> A: Could the reviewer leverage more why only one model can be chosen at deployment? We conducted the experiments following the state-of-the-art work in connecting DP and certified robustness bounds [4, 5].
>
> Q3: The attacks used for evaluating the empirical adversarial robustness are too weak. Only 10 steps are used. More iterative steps, e.g., 1000 or 10000 step PGD I-FGSM, should be provided in Figs. 4 and 5.
>
> A: We do agree that 1000 or 10000-step attacks can be used. However, it may not be necessary in the first place, since with only 10-step attacks, we have clearly exposed severe vulnerabilities of ``DP deep neural networks'' under adversarial examples, especially with rigorous privacy protection ($\epsilon = 0.2$). In fact, it always is challenging to achieve high model utility under rigorous DP protection. With 10-step attacks, we also demonstrated that addressing the trade-off between privacy loss (preserving DP to protect the training data) and certified robustness is a non-trivial and largely open task, given tight privacy budgets. Considering that this paper is the first paper introducing the problem of preserving DP in adversarial learning, 10-step attacks have already shown potential and critical research problem for exploring better trade-off among privacy loss, model utility, and certified robustness under attacks. Note that, we focus on DP deep neural networks, not clean models as in [1, 2].
>
> We are running the attacks with up to 100 step-attacks in Figs. 4 and 5. We expect that the results do still hold. The results will be added later.

---

> ### Author Response · Authors · 2019-11-09
> **Response to Review 1 - Part 2**
>
> Q4: The results of the proposed algorithm in Figure 7 is much better than the state-of-the-art, could the authors explain clearly where such improvements come from?
>
> A: There is a misunderstanding here.
> First, Note that the experiments on clean models in [1, 2] and our experiment in Fig. 7 are completely different. In this paper, we focus on improving the certified robustness of "DP deep neural networks" (not clean models).
>
> Second, in [1, 2], clean ResNet models, which have significantly larger latent spaces (easier to attack), are tested under much stronger attacks (DeepFool or 100 to 1000-step PGD). It is natural to register a drop in certified accuracy in that scenarios, compared to our 3-step attacks (weaker attacks) and our CNN models (smaller latent space, easier to defend) under DP preservation in CIFAR10. It is unfair to compare side-by-side these two settings.
>
> Third, the key advantage in our mechanism compared with PixelDP (and also with [1, 2]) is the Composition Robustness Bound, which is derived by randomizing both input space and latent space (the first hidden layer) together. That leverages the capacity of both input space and latent space to derive certified robustness bound against adversarial examples. In [1, 2], only the input space was randomized. Randomizing both input space and latent space can significantly improve the robustness bounds in DP deep neural networks. Note that adapting existing defensive mechanisms, such as [1, 2], into DP preserving to protect the training data can only be applied with great care, since robustness can incur privacy risk [3].
> The second advantage in our mechanism is that our training algorithm is an ensemble adversarial training approach, while [2] simply follows the training algorithm of PixelDP, which is not adversarial training. [1] is also not adversarial training. These two key advantages make our mechanism robust to adversarial examples under DP protections.
>
> Finally, to completely clear the reviewer's doubts about our experimental results, the anonymized code of our mechanism is available at "https://www.dropbox.com/s/1a536ghyadsbknt/ICLR2020.zip?dl=0" (active during the discussion phase). Please feel free to run stronger attacks against our mechanism and compare it with [1] and [2], even the comparison may not be totally fair between a DP model and clean models.
>
> P/s: It is also important to note that $x \in [-1, 1]^d$ in our setting, which is different from a common setting, $x \in [0, 1]^d$. Thus, a given attack size $\mu_a = 8/255$ in the setting of $x \in [0, 1]^d$ is equivalent to an attack size $2\mu_a = 16/255$ in our setting. The reason for using $x \in [-1, 1]^d$ is to achieve better model utility, while retaining the same global sensitivities to preserve DP, compared with $x \in [0, 1]^d$. This was clearly stated in our experimental setting.
>
> [1] "Filling the Soap Bubbles: Efficient Black-Box Adversarial Certification with Non-GaussianSmoothing," https://openreview.net/forum?id=Skg8gJBFvr.
> [2]  Salman,  Hadi,  et  al.   "Provably  Robust  Deep  Learning  via  Adversarially  Trained  SmoothedClassifiers." NeurIPS (2019).
> [3] Song, Liwei and Shokri, Reza and Mittal, Prateek. "Privacy Risks of Securing Machine Learn-ing Models against Adversarial Examples."  Proceedings of the 2019 ACM SIGSAC Conference onComputer and Communications Security - CCS ’19.
> [4] Mathias Lecuyer, Vaggelis Atlidakis, Roxana Geambasu, Daniel Hsu, and Suman Jana.  Certified robustness to adversarial examples with differential privacy. In S&P, 2019.
> [5] NhatHai Phan, Minh N. Vu, Yang Liu, Ruoming Jin, Dejing Dou, Xintao Wu, and My T. Thai. Heterogeneous gaussian mechanism: Preserving differential privacy in deep learning with provable robustness. In IJCAI, 2019.

---

### Official Review · AnonReviewer2 · 2019-10-26
**Official Blind Review #2**

**Rating:** 6

**Review:**

In this submission, the authors address a challenging task, in which the goals are three-fold: 1) preserve the privacy of training data in terms of DP; 2) both provably and practically robust to adversarial examples; and 3) maintain high model utility. The authors demonstrate the advantage of the proposed method both theoretically and experimentally.

Overall, this is a solid work. However, I have the following concerns. If the authors can clarify them during the rebuttal, I am willing to increase my review score.

1) On page 9, the authors mentioned several existing DP-preserving algorithms with provable robustness. Could the authors clearly state the novelty of the proposed method given these existing methods? Thus it would be clear to access the technique contribution of this submission.

2) How about the scalability of the proposed method when applying to real-world scenarios? In the experiment part, the authors show the results on MNIST and CIFAR-10; however, the scale of these two datasets are kind of small compared to real-world applications. Can the authors experimentally show the scalability of the proposed method by increasing the number of samples?

3) This question will not be counted towards the review: Do the authors try other datasets? MNIST and CIFAR-10 are relatively simple image datasets, how about real images? How about non-image datasets?

4) This submission is longer than 8 pages, it is OK. But the main results are actually in appendix, and the reviewers are forced to read more than 10 pages. It would be better to adjust the contents and have a balanced organization.

A good thing to mention is: It is great to see that the authors conduct experiment comparison and analysis with a relatively small value of epsilon (i.e., epsilon = 0.2).

**Experience Assessment:**

I have published one or two papers in this area.

**Review Assessment: Checking Correctness Of Derivations And Theory:**

I assessed the sensibility of the derivations and theory.

**Review Assessment: Checking Correctness Of Experiments:**

I carefully checked the experiments.

**Review Assessment: Thoroughness In Paper Reading:**

I read the paper at least twice and used my best judgement in assessing the paper.

---

> ### Author Response · Authors · 2019-11-09
> **Response to Review 2**
>
> We thank all reviewers for the constructive feedback. We have thoroughly addressed all the questions raised in our following responses as follows.
>
> Q1: Could the authors clearly state the novelty of the proposed method given existing DP-preserving algorithms with provable robustness?
>
> A: The novelty of our proposed approach compared with SecureSGD (Phan et al., IJCAI'19: Heterogeneous gaussian mechanism...), which is the first and the only DP-preserving algorithm with provable robustness, is in four folds: (1) A new mechanism which theoretically and practically integrates ensemble adversarial learning into DP preservation to protect the training data, through the new concept of DP adversarial examples. While, SecureSGD is not an adversarial training algorithm; (2) The robustness bounds are significantly strengthened by the Composition Robustness Theory, in which we leverage both randomized input space and randomized latent space (the first hidden layer), as independent defensive mechanisms, to derive the robustness bound. This is critical to improve the robustness of DP deep neural networks, under the same privacy loss. While, SecureSGD does not have this ability; (3) Our mechanism offers a new and more comprehensive tool to explore the unknown correlations among DP preserving, adversarial learning, and certified robustness. While, SecureSGD does not offer this; and (4) A more extensive experiment was conducted in our study, offering a more insightful picture to the unknown trade-off among DP preserving, adversarial learning, and certified robustness.
>
> Q2: How about the scalability of the proposed method when applying to real-world scenarios?
>
> A: Scalability is a common and non-trivial issue in DP deep neural networks, and especially with adversarial learning using images, since the bottleneck mainly from the limited computational efficiency in generating adversarial examples. In fact, we need to generate adversarial examples for each training step. Unfortunately, with the state-of-the-art adversarial example crafting algorithms, generating such a large number of adversarial examples at each step for DP Adversarial Training is costly in terms of time and computation complexity. In addition, the convergence speed in DP models usually is slower than clean and non-DP training algorithms. All of these problems pose an issue to scale DP Adversarial Learning.
>
> One promising solution to address the scalability issue is applying our DP preserving mechanism on top of scalable pre-trained models, which were trained on public datasets; thus improving the convergent speed and scalability. We are working on this issue; however, as we mentioned in our conclusion, scalable DP Adversarial Training is beyond the scope of this paper. In this paper, we focus on establishing a solid theoretical foundation for DP preservation in adversarial learning, and strengthening the certified robustness bounds (under DP protection). The good thing is that our mechanism is not restricted to the type of activation functions. Therefore, our mechanism has a great potential to be scaled up.
>
> Q3: Can the authors experimentally show the scalability of the proposed method by increasing the number of samples?
>
> A: As mentioned in Q2, we need to address the scalability in several directions: (1) We need new and fast algorithms to generate a large number of adversarial examples crafted by a large number of iterative steps. Given the computational efficiency bottleneck in generating adversarial examples, in a short time period of the rebuttal, it would be challenging for us to get a reliable result by empirically running our mechanism with large-scale datasets and large-scale networks; and (2) We need to carefully apply our mechanism on top of a scalable pre-trained model. We are working on this, but it requires a significant amount of effort to get reliable and practical results. We should not expect to solve every issue in just one paper, since this is an open problem requiring a long-run effort.
>
> Q4: Do the authors try other datasets? How about non-image datasets?
>
> A: CIFAR10 is a simple image dataset, but it is sufficiently difficult for DP deep neural networks. It is challenging to achieve high model utility under rigorous DP protection ($\epsilon \leq 2$). Our mechanism applied on ResNet18 for CIFAR100 works well. Regarding non-image datasets, our mechanism can be applied, as long as we can generate adversarial examples. We have tried our mechanism on a health care dataset, and it works well. We are also in the middle of testing our mechanism on sensitive textual datasets, based upon CRNN models. Although we are not ready to report the results on these datasets, this demonstrates the ability of our mechanism to be applied in a variety of domain applications.
>
> Q5: It would be better to adjust the contents and have a balanced organization.
>
> A: We will balance our paper to include experimental results in the main body of our revision version.

---

### Author Response · Authors · 2019-11-09
**Summary of Changes in the Revision**

Dear All Reviewers,

We thank all reviewers for the constructive feedback. We have thoroughly addressed all the questions raised in our following responses. The summary of changes is as follows:

- We have reordered the Lemma 3 and Theorem 1. The Theorem 1 has been revised to accurately compute the privacy budget for learning parameters $\theta_1$.
- We have added the Approximation Error Bounds in Appendix N.

We do appreciate valuable discussions and efforts from the reviewers. We look forward to feedback from the reviewers regarding our rebuttal.

Best regards,

---

### Decision · Program_Chairs · 2019-12-19

**Decision:**

Reject

**Comment:**

The authors propose a framework for relating adversarial robustness, privacy and utility and show how one can train models to simultaneously attain these properties. The paper also makes interesting connections between the DP literature and the robustness literature thereby porting over composition theorems to this new setting.

The paper makes very interesting contributions, but a few key points require some improvement:
1) The initial version of the paper relied on an approximation of the objective function in order to obtain DP guarantees. While the authors clarified how the approximation impacts model performance in the rebuttal and revision, the reviewers still had concerns about the utility-privacy-robustness tradeoff achieved by the algorithm.

2) The presentation of the paper seems tailored to audiences familiar with DP and is not easy for a broader audience to follow.

Despite this limitations, the paper does make significant novel contributions on an improtant problem (simultaneously achieveing privacy, robustness and utility) and could be of interest.

Overall, I consider this paper borderline and vote for rejection, but strongly encourage the authors to improve the paper wrt the above concerns and resubmit to a future venue.